# No Experts, No Problem: Avoidance Learning from Bad Demonstrations

**Huy Hoang**
Singapore Management University
Institute for Infocomm Research, A*STAR, Singapore
mh.hoang.2024@phdcs.smu.edu.sg

**Tien Mai**
Singapore Management University
atmai@smu.edu.sg

**Pradeep Varakantham**
Singapore Management University
pradeepv@smu.edu.sg

## Abstract

This paper addresses the problem of learning avoidance behavior within the context of offline imitation learning. In contrast to conventional methodologies that prioritize the replication of expert or near-expert demonstrations, our work investigates a setting where expert (or desirable) data is absent, and the objective is to learn to eschew undesirable actions by leveraging demonstrations of such behavior (i.e., learning from negative examples).

To address this challenge, we propose a novel training objective grounded in the maximum entropy principle. We further characterize the fundamental properties of this objective function, reformulating the learning process as a cooperative inverse Q-learning task. Moreover, we introduce an efficient strategy for the integration of unlabeled data (i.e., data of indeterminate quality) to facilitate unbiased and practical offline training. The efficacy of our method is evaluated across standard benchmark environments, where it consistently outperforms state-of-the-art baselines.

## 1 Introduction

Imitation learning [1, 47, 23] offers a feasible alternative to Reinforcement Learning (RL), enabling agents to learn directly from expert demonstrations without the need for explicit reward signals. It has proven effective in several tasks, even with limited expert data, and is particularly useful in capturing human preferences.

Most existing imitation learning approaches prioritize maximizing task performance (i.e., expected return) by closely mimicking expert demonstrations [18, 11, 28, 12]. However, in practice, expert or near-expert demonstrations may be unavailable or insufficient. First, acquiring expert demonstrations can be prohibitively expensive or time-consuming, as it often requires specialized skills or resources [40].

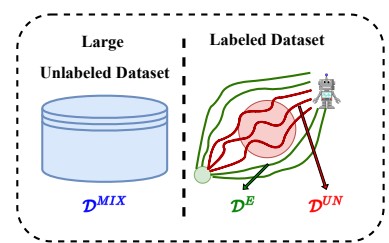

Figure 1: Datasets for offline imitation learning. In our setting, we only use Undesirable labeled dataset ($\mathcal{D}^{\text{UN}}$) and Unlabeled dataset ($\mathcal{D}^{\text{MIX}}$).

Second, in many domains, such as healthcare [15] or autonomous driving [6], expert-level performance may be rare or difficult to define, leading to a scarcity of high-quality demonstrations [30].

Third, expert demonstrations may not cover the full range of scenarios or edge cases, limiting their generalizability [43].

Instead of expert demonstrations, there may be collections of undesirable demonstrations that should be avoided and sub-optimal behaviors that may be partially imitated. In the development of self-driving cars [5], while companies may collect user driving data to train their models, the system must ensure it does not replicate faulty behavior, such as traffic violations or unsafe driving practices [6]. In the field of treatment optimization, data may include actions that led to bad patient outcomes, and the system must learn to avoid such behaviors [15].

Although this is an important and interesting problem setting, there has been limited research addressing it effectively. Formally, given a dataset containing *undesired demonstrations* we wish to avoid, alongside a much larger unlabeled dataset consisting of both desired and undesired demonstrations, the goal is to learn a desirable policy that avoids undesired trajectories. To the best of our knowledge, only SafeDICE [20] directly addresses this setting by mixing the undesirable and unlabeled datasets, assigning negative weights to the former, and then mimicking the mixed policy. This approach, however, may suffer from the fact that *the mixed policy is not necessarily a desirable one* to follow, which is often the case in practice. There are existing methods, such as preference-based RL [8, 32, 17] and Discriminator-Weighted Behavioral Cloning (DWBC) [45] that, while not specifically designed to address this problem, can be adapted to handle it.

Towards addressing this problem of avoiding undesirable trajectories, our key contributions are:

- First, we formulate the new learning problem as a cooperative training task to ensure stable training, in contrast to prior imitation learning approaches where the objective is adversarial [18, 11, 28].

- To efficiently solve the training problem using limited undesirable demonstrations, we introduce our algorithm, *UNIQ*, which employs an occupancy correction mechanism to recast the training objective, allowing the expectation over the undesirable policy to be unbiasedly approximated using unlabeled trajectories.

- We evaluate our method on two popular benchmark environments, Mujoco and Safety-Gym, using public datasets [38, 21]. Our experiments demonstrate superior performance compared to several state-of-the-art baselines.

## 2   Related Work

**Imitation learning.** Imitation learning is a key technique for learning from demonstrations. Behavioral Cloning (BC) maximizes the likelihood of expert demonstrations but often struggles due to distributional shift [40]. To improve, Generative Adversarial Imitation Learning [18, 11] aligns the learner's policy with the expert's using GANs [14], while SQIL [39] assigns simple rewards to expert and non-expert demonstrations to learn a value function. PWIL [9] uses the Wasserstein distance [44] to compute rewards. While these methods show promise, they rely on online interaction, which can be impractical. For offline learning, AlgaeDICE [34] and ValueDICE [29] use Stationary Distribution Correction Estimation (DICE) but face stability issues. Inspired by ValueDICE, O-NAIL [4] introduced an offline method without adversarial training. IQ-learn [12], a popular approach, supports both online and offline learning and offers a state-of-the-art framework with several variants developed based on it [2, 19]. Unlike the above mentioned works, our focus in this paper is on offline **reverse** imitation learning that aims to avoid undesirable trajectories, as opposed to imitating expert trajectories. As indicated earlier, this requires fundamentally different methods due to the nature of the problem.

**Imitation Learning from Sub-optimal Demonstrations.** There are two main research directions in this area. The first focuses on online and offline preference-based imitation learning methods. Online approaches, such as T-REX [8], PrefPPO [31], and PEBBLE [32], leverage ranked sub-optimal demonstrations to learn a preference-based reward function using the Bradley-Terry model [7]. While these methods achieve strong performance, they rely on interactions with the environment. In contrast, offline methods, such as those proposed by [24, 22, 17], rely heavily on an extensive dataset of pairwise trajectory comparisons. SPRINQL [19] addressed this reliance on a large offline dataset, by utilizing demonstrations categorized into different levels of expertise, resulting in better performance with fewer comparison data points. While these methods concentrate on imitating

preferred (or expert) trajectories, our focus is on avoiding non-preferred (or undesired) trajectories. This important distinction necessitates solving a different optimization problem and consequently a change in methodology.

The second direction focuses the use of additional unlabeled datasets to enhance learning from expert data. Beginning with DemoDICE [26], several DICE-based methods [33, 25, 46] have been developed to utilize small sets of expert demonstrations, supplemented by larger unlabeled datasets. In addition to these DICE-based methods, DWBC [45] proposes a simple and efficient method based on training a classifier using positive-unlabeled learning [27]. SafeDICE [20] presents a DICE-based framework capable of learning from undesirable demonstrations. This method combines an undesirable policy (represented by an undesirable dataset) with a random policy (represented by a larger unlabeled dataset), assigns negative weights to the undesirable policy, and then applies a standard DICE-based approach [26, 25] to mimic the combined policy by minimizing the KL divergence between the learning policy and the mixed policy. The primary limitation of this approach is that, with the quality of the unlabeled dataset being unknown, imitating the mixed policy may not lead to the desired learning outcome.

## 3 Background

**Preliminaries.** We consider a MDP defined by the following tuple $\mathcal{M} = \langle S, A, r, P, \gamma, s_0 \rangle$, where $S$ denotes the set of states, $s_0$ represents the initial state set, $A$ is the set of actions, $r : S \times A \to \mathbb{R}$ defines the reward function for each state-action pair, and $P : S \times A \to S$ is the transition function, i.e., $P(s'|s, a)$ is the probability of reaching state $s' \in S$ when action $a \in A$ is made at state $s \in S$, and $\gamma$ is the discount factor. In reinforcement learning (RL), the aim is to find a policy that maximizes the expected long-term accumulated reward: $\max_\pi \left\{ \mathbb{E}_{(s,a) \sim \rho_\pi} [r(s, a)] \right\}$, where $\rho_\pi$ is the occupancy measure of policy $\pi$: $\rho_\pi(s, a) = (1 - \gamma)\pi(a|s) \sum_{t=1}^{\infty} \gamma^t P(s_t = s|\pi)$.

**Offline MaxEnt IRL** The goal of offline MaxEnt IRL is to derive a reward function $r(s, a)$ based on a set of expert demonstrations, $\mathcal{D}^E$ without interacting with the environment. Let $\rho^E$ denote the occupancy measure of the expert policy. The MaxEnt IRL framework [47, 16], aims to recover the expert's reward function by maximizing the gap between the expected reward under the expert's policy and the maximum expected reward across all other policies (as determined by the inner minimization).:

$$\max_r \min_\pi \left\{ \mathbb{E}_{\rho^E}[r(s, a)] - \mathbb{E}_{\rho_\pi}[r(s, a)] - H(\pi, \mu) - \psi(r) \right\} \tag{1}$$

where $H(\pi, \mu) = \mathbb{E}_{\rho^\pi} \left[ -\beta \log \frac{\pi(s,a)}{\mu(a|s)} \right]$ is the discounted causal entropy of the policy $\pi$ given a behavior policy $\mu$, $\beta$ is the regularization strength, and $\psi(r) : \mathbb{R}^{S \times A} \to \mathbb{R}$ is a convex reward regularizer. When $\mu$ is uniform, the objective reduces to the original MaxEnt objective [47]. In the offline RL setting, $\mu$ is set to be the behavior policy that generated the offline dataset. As a result, this objective imposes a conservative KL constraint on the learned policy, ensuring it stays close to the behavior policy. This helps mitigate the out-of-distribution issues that commonly arise in offline RL [36, 16].

**Inverse Q-learning (IQ-Learn) from expert demonstrations.** Given a reward function $r$ and a policy $\pi$, the soft Bellman equation is defined as

$$\mathcal{B}_r^\pi[Q](s, a) = r(s, a) + \gamma \mathbb{E}_{s'}[V^\pi(s')], \text{ where } V^\pi(s) = \mathbb{E}_{a \sim \pi(a|s)} \left[ Q(s, a) - \beta \log \frac{\pi(a|s)}{\mu(a|s)} \right].$$

The Bellman equation $\mathcal{B}_r^\pi[Q] = Q$ is contractive and always yields a unique Q solution [12, 36]. In IQ-Learn [12], they further define an inverse soft-Q Bellman operator $\mathcal{T}^\pi[Q] = Q(s, a) - \gamma \mathbb{E}_{s'}[V^\pi(s')]$. [12] show that for any reward function $r(a, s)$, there is a unique $Q^*$ function such that $\mathcal{B}_r^\pi[Q^*] = Q^*$, and for a $Q^*$ function in the $Q$-space, there is a unique reward function $r$ such that $r = \mathcal{T}^\pi[Q^*]$. This result suggests that one can safely transform the objective function of Equation 1 from $r$-space to the $Q$-space as follows:

$$\max_Q \min_\pi \quad \Phi(\pi, Q) = \mathbb{E}_{\rho^E}[\mathcal{T}^\pi[Q](s, a))] - \mathbb{E}_{\rho_\pi}[\mathcal{T}^\pi[Q](s, a)] - H(\pi, \mu) - \psi(\mathcal{T}^\pi[Q](s, a))) \tag{2}$$

which has several advantages, namely, the objective function $\Phi(\pi, Q)$ is concave in $\pi$ and convex in Q. Moreover, the inner problem $\min_\pi \Phi(\pi, Q)$ has a closed form solution as $\exp(Q(s,a)/\beta)/\sum_a \exp(Q(s,a')/\beta)$ As a result, the maximin problem can be converted to a *non-adversarial* problem in the Q-space as:

$$\max_Q \; \mathbb{E}_{\rho^E}[\mathcal{T}[Q](s,a))] - (1-\gamma)\mathbb{E}_{s_0}[V^Q(s)] - \psi(\mathcal{T}[Q](s,a))) \tag{3}$$

where $\mathcal{T}[Q](s,a)) = Q(s,a) - \gamma\mathbb{E}_{s'\sim P(s'|s,a)}[V^Q(s')]$ and $V^Q(s) = \beta\log\left(\sum_a \mu(a|s)\exp(Q(s,a)/\beta)\right)$, which is a softmax of the Q function. The reward function can then be recovered as $r^Q(s,a) = \mathcal{T}[Q](s,a)$. Thus, in (3), the objective can be interpreted as training a reward function (via a Q-function) that maximizes the expected reward under the expert policy while minimizing the overall expected reward. An important and appealing characteristic of the inverse Q-learning framework described above is that the learning objective is concave in Q, theoretically guaranteeing convergence to a unique solution in the Q-space. This property also ensures stable and robust training outcomes in practice. Later, we will show that this property does not hold in our learning-from-mistakes setting, prompting us to adopt a more recent framework, Extreme Q-learning [13], to reformulate the training process using updates based on convex loss functions.

## 4 Undesired Demonstrations driven Inverse Q-learning - UNIQ

We now describe our approach for avoidance (and not imitation) learning of undesirable demonstrations in the presence of an unlabeled (unknown quality) demonstrations dataset. First, we explain our novel MaxEnt objective that facilitates this avoidance learning problem (Section 4.1). However, directly solving this novel objective in reward space is inefficient, so we next provide a reformulation in the the Q-space (Section 4.2). Third, we leverage the extreme Q-learning method (Section 4.3) to solve this reformulated objective. Finally, we make our approach sample efficient by utilizing an unlabeled dataset of trajectories (Section 4.4).

### 4.1 Novel MaxEnt Objective

In our setting, we have a set of undesired demonstrations, denoted as $\mathcal{D}^{\mathrm{UN}}$, along with a supplementary set of unlabeled demonstrations, denoted as $\mathcal{D}^{\mathrm{MIX}}$. The unlabeled dataset $\mathcal{D}^{\mathrm{MIX}}$ may contain a mix of random, undesired, and expert demonstrations, and it will be used to support offline learning. Let $\rho^{\mathrm{UN}}$ be the occupancy measure (or stationary distribution) of the undesired policy (represented by the undesired dataset).

Adapting the MaxEnt RL framework, the goal here is to find a reward function and a policy that minimizes: *(a) the expected reward obtained by undesirable state, action pairs (part of undesirable trajectories); and (b) negative of the expected reward obtained by state, action pairs not in undesirable trajectories.* To this end, we introduce the following new learning objective:

$$\min_r \min_\pi \left\{ L(\pi, r) = \mathbb{E}_{\rho^{\mathrm{UN}}}[r(s,a)] - \mathbb{E}_{\rho_\pi}[r(s,a)] - H(\pi, \mu) + \psi(r) \right\} \tag{4}$$

In our context, where the objective contrasts with the standard learning-from-expert-demonstration scheme, the learning problem is *no longer adversarial* as in prior imitation learning approaches [18, 28]. Instead, it can be framed as a *cooperative learning problem*, where the objective is to jointly identify a policy and reward function that minimize the objective function $L(\pi, r)$. Solving (4) directly encourages the learning policy to deviate as much as possible from the undesired policy, which is derived from undesirable demonstrations (we give a detailed discussion in appendix). This approach contrasts with that of SafeDICE [20], which minimizes the KL divergence between the learning policy and the mixed policy, which could suffer from low quality mixing datasets.

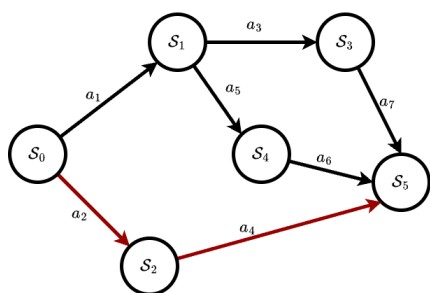

Figure 2: An illustrative MDP.

**Illustrative Example**   We provide an illustrative example to demonstrate how the proposed objective guides the learning of a good policy from undesirable demonstrations. Consider the simple MDP shown in Figure 2. There are three trajectories from $S_0$ to $S_5$: $p_1 = [S_0, S_2, S_5]$, $p_2 = [S_0, S_1, S_4, S_5]$, and $p_3 = [S_0, S_1, S_3, S_5]$. We treat $p_1$ as an undesirable trajectory, while $p_2$ and $p_3$ are considered optimal. The undesirable dataset is $\mathcal{D}_{\text{und}} = \{p_1\}$, and the unlabeled dataset includes all trajectories: $\mathcal{D}_{\text{unlabeled}} = \{p_1, p_2, p_3\}$. Under Equation (4), the training objective becomes:

$$\min_r \min_\pi \left\{ R(p_1) - (P^\pi(p_1)R(p_1) + P^\pi(p_2)R(p_2) + P^\pi(p_3)R(p_3)) + \mathcal{H}(\pi) \right\},$$

where $R(p)$ denotes the total accumulated reward along trajectory $p$, e.g., $R(p_1) = r(S_0, a_1) + r(S_1, a_3) + r(S_3, a_7)$ (ignoring discount factor $\gamma$ for simplicity), $P^\pi(p)$ is the probability of trajectory $p$ under policy $\pi$, and $\mathcal{H}(\pi)$ is the entropy regularization term. Simplifying the objective yields:

$$-P^\pi(p_2)R(p_2) - P^\pi(p_3)R(p_3) + (1 - P^\pi(p_1))R(p_1).$$

To minimize this expression, the algorithm is incentivized to assign higher rewards to $p_2$ and $p_3$, and a lower reward to $p_1$. Consequently, to minimize the expected reward (note the negative coefficients), the optimal policy $\pi$ should assign high probabilities to $p_2$ and $p_3$, and a low probability to $p_1$. This encourages the policy to avoid undesirable behaviors and prefer optimal ones—even in the absence of direct expert demonstrations.

## 4.2   Learning in the Q-space

Directly solving Eq. 4 is not efficient. Instead, prior research indicates that transforming the reward learning problem into the Q-space and simplifying it can improve efficiency [12, 17]. As discussed in Section 3, there is a one-to-one mapping between any reward function $r$ and a corresponding function $Q$ in the Q-space. Therefore, the minimization problem in Eq. 4 can equivalently be transformed as:

$$\min_Q \min_\pi \left\{ L(\pi, Q) = \mathbb{E}_{\rho^{\text{UN}}}[\mathcal{T}^\pi[Q](s,a))] - \mathbb{E}_{\rho_\pi}[\mathcal{T}^\pi[Q](s,a) - H(\pi) + \psi\left(\mathcal{T}^\pi[Q](s,a)\right) \right\} \quad (5)$$

where $\mathcal{T}^\pi[Q](s,a)) = Q(s,a) - \gamma\mathbb{E}_{s'}[V^\pi(s')]$ and $V^\pi(s) = \mathbb{E}_{a\sim\pi(a|s)}[Q(s,a) - \beta\log\frac{\pi(a|s)}{\mu(a|s)}]$. Compared to the primary objective of the standard MaxEnt framework [18, 12], our objective function in Eq. 5 is no longer adversarial with respect to $Q$ and $\pi$.

Before we describe ways of simplifying objective in (5), we wish to highlight the properties that make IQ-learn and its variants efficient and effective:
**Property 1**: Closed-form optimization over $\pi$.
**Property 2**: Concavity of the simplified objective in the Q-space.

In comparison to imitation learning problem in the MaxEnt framework, since we negate the objective function, it alters its nature, turning a concave function into a convex one. We show that despite this change, property 1 is still preserved in the new objective. However, the simplified objective obtained by using property 1 is neither convex nor concave.

We first focus on **Property 1**, where we provide a closed form expression for policy, $\pi^Q$. If the regularizer function $\psi(\cdot)$ is non-decreasing, then the objective function $L(\pi, Q)$ is convex in $\pi$. Furthermore, the minimization problem $\min_\pi L(\pi, Q)$ retains a closed-form solution, thereby simplifying the learning objective as shown in Proposition 4.1.

**Proposition 4.1.** *The function $L(\pi, Q)$ is convex in $\pi$ and the problem $\min_\pi L(\pi, Q)$ has a unique optimal solution at $\pi^Q(s, a) = \frac{\exp(Q(s,a)/\beta)}{\sum_{a'} \exp(Q(s,a')/\beta)}$. Moreover, the learning objective function can be simplified as:* $\min_Q \min_\pi \{L(\pi, Q)\} = \min_Q \mathcal{F}(Q)$, *where*

$$\mathcal{F}(Q) = \mathbb{E}_{\rho^{\text{UN}}}[r^Q(s,a)] - (1-\gamma)\mathbb{E}_{s_0}[V^Q(s_0)] + \psi(r^Q),$$

$V^Q(s) = \beta\log\left(\sum_a \mu(a|s)\exp(Q(s,a)/\beta)\right)$ *and* $r^Q(s,a) = Q(s,a) - \gamma\mathbb{E}_{s'\sim P(\cdot|s,a)}V^Q(s')$.

Proposition 4.1 shows that, similar to standard inverse Q-learning algorithms, our training objective can be framed as an optimization problem over the Q-space, where the optimal policy can be computed as the soft-max of the Q-function. Thereby preserving **Property 1**.

However with regards to **Property 2**, while in inverse Q-learning with expert demonstrations, the training objective is convex within the Q-space, this is not the case in our context, where the objective $\mathcal{F}(Q)$ is neither convex nor concave in $Q$. We state this observation as follows:

**Proposition 4.2.** *The learning objective function $F(Q)$ is not convex in $Q$.*

This non-convexity arises from the presence of the term $-V^Q$ in the objective function, where $V^Q$ is a log-sum-exp of $Q$, which is itself convex in $Q$. This directly implies that $F(Q)$ is not convex in $Q$. Such non-convexity is a significant disadvantage compared to prior inverse Q-learning approaches that learn from expert demonstrations. In the following, we leverage the Extreme Q-learning method [13] to reformulate the training as updates based on convex loss functions, effectively mitigating the challenges associated with non-convexity.

### 4.3 Extreme Q-learning and Convexity

In the above formulations, the optimal value function $V^Q$ can be expressed as a log-sum-exp of the Q-function, i.e., $V^Q(s) = \beta \log(\sum_a \mu(a|s) \exp(Q(s,a)/\beta))$. This log-sum-exp computation is often impractical as $\mu(a|s)$ is not explicitly known and the action space would be large. Furthermore, as discussed earlier, it is the primary cause of the non-convexity in our learning objective function. To address this issue, we leverage the Extreme Q-Learning (XQL) method [13] to update $V$ simultaneously with $Q$-function. Specifically, we define the Extreme-V function as: $J(V|Q) = \mathbb{E}_{(s,a)\sim\mu} e^t - (t) - 1$, where $t = \frac{Q(s,a)-V(s)}{\beta}$ where it can be shown that minimizing $J(V|Q)$ over $V$ yields an optimal solution $V^*(s)$ such that: $V^*(s) = \beta \log\left(\sum_a \mu(a|s)e^{Q(s,a)/\beta}\right)$. We then define the Q-learning function conditional on $V$ as $\widetilde{F}(Q|V)$, which is defined as

$$\widetilde{F}(Q|V) = \mathbb{E}_{\rho^{\text{UN}}}\left[Q(s,a) - \gamma\mathbb{E}_{s'}[V(s')]\right] - (1-\gamma)\mathbb{E}_{s_0}[V(s_0)] + \psi(Q(s,a) - \gamma\mathbb{E}_{s'}[V(s')]).$$

The minimization of our learning objective $F(Q)$ can be achieved through the following alternating updates: (i) **Update $Q$:** Minimize the objective $\widetilde{F}(Q|V)$, and (ii) **Update $V$:** Minimize the Extreme-V function $J(V|Q)$. This procedure is particularly convenient as both updates can be shown to have favorable convexity properties. Specifically, we demonstrate that the loss functions are convex in their respective spaces:

**Proposition 4.3.** *Under any convex regularizer $\psi$, both $\widetilde{F}(Q|V)$ and $J(V|Q)$ are **convex** in $Q$ and $V$, respectively.*

### 4.4 Unbiased Q-Learning with Unlabeled Data

We now discuss the optimization in the Q-space. Directly solving $\min_Q \widetilde{F}(Q|V)$ using only the undesirable dataset would be inefficient due to its small size (as shown in Appendix E.5). Furthermore, directly combining the undesirable and unlabeled datasets for training is risky, as the unknown quality of the unlabeled dataset could lead to learning in the wrong direction. For instance, the model could unintentionally mimic low-quality demonstrations, as might occur in methods like SafeDICE. To overcome this challenge, we address the following critical question: ***How can we unbiasedly estimate the objective $\min_Q \widetilde{F}(Q|V)$ while effectively utilizing the unlabeled dataset?***

To address this question, we leverage the larger set of unlabeled data $\mathcal{D}^{\text{MIX}}$ to enhance the offline training. To achieve this, we first let $\rho^{\text{MIX}}$ be the occupancy measure (or stationary distribution) of the policy represented by unlabeled dataset. We rewrite the expectation over $\rho^{\text{UN}}$ as: $\mathbb{E}_{\rho^{\text{UN}}}[r^Q(s,a)] = \mathbb{E}_{\rho^{\text{MIX}}}[\tau(s,a)r^Q(s,a)]$, where $\tau(s,a) = \frac{\rho^{\text{UN}}(s,a)}{\rho^{\text{MIX}}(s,a)}$ represents the occupancy ratio between $\rho^{\text{UN}}$ and $\rho^{\text{MIX}}$, we then rewrite the learning objective as follows:

$$\min_Q \left\{\widetilde{F}(Q|V) = \mathbb{E}_{\rho^{\text{MIX}}}[\tau(s,a)r^Q(s,a)] - (1-\gamma)\mathbb{E}_{s_0}[V(s_0)] + \psi(r^Q)\right\} \tag{6}$$

where $r^Q(s,a) = Q(s,a) - \gamma\mathbb{E}_{s'}[V(s')]$. In this approach, the expectation $\mathbb{E}_{\rho^{\text{MIX}}}[\tau(s,a)r^Q(s,a)]$ can be empirically approximated using the unlabeled samples from $\mathcal{D}^{\text{MIX}}$, where $\tau(s,a)$ acts as an occupancy correction. This correction allows us to leverage samples from the unlabeled dataset to *unbiasedly* estimate the expectation over the undesirable policy. A key challenge here is that the occupancy ratio $\tau(s,a)$ is unknown. To address this, we propose estimating the ratio by solving the following implicit maximization problem:

$$\max_{\vartheta:S\times A\to[0,1]} \{g(\vartheta)\} \tag{7}$$

where $g(\vartheta) = \mathbb{E}_{\rho^{\text{MIX}}}[\log(1 - \vartheta(s,a)] + \mathbb{E}_{\rho^{\text{UN}}}[\log(\vartheta(s,a))]$. The above formulation is similar to the discriminator-based formulations widely used in prior adversarial imitation learning work [18, 23]. In fact, the objective function $g(\vartheta)$ is strictly concave in $\vartheta$. Let $\vartheta^*$ denote the unique optimal solution to (7), then the occupancy ratio can be estimated as $\tau(s,a) = \frac{\vartheta^*(s,a)}{1 - \vartheta^*(s,a)}$.

So, our learning process can be broken down into two steps. In the first step, we learn the occupancy ratios by solving the maximization problems presented in 7. Following this, we optimize the following problem to learn $Q$ and $V$ functions.

$$\min_Q \left\{ \widetilde{F}(Q|V) = \mathbb{E}_{\rho^{\text{MIX}}}\left[ \frac{\vartheta^*(s,a)}{1 - \vartheta^*(s,a)} r^Q(s,a) \right] - (1-\gamma)\mathbb{E}_{s_0}[V(s_0)] + \psi(r^Q) \right\} \tag{8}$$

It is important to note that we utilize the stationary distribution $\rho^{\text{MIX}}$ (represented by trajectories in the unlabeled dataset) in the objective function in (8). However, thanks to the occupancy correction $\frac{\vartheta^*(s,a)}{1 - \vartheta^*(s,a)}$, the outcome of the training is theoretically independent of the quality of the unlabeled policy $\rho^{\text{MIX}}$. This distinguishes our approach from SafeDICE [20], where the performance heavily relies on the quality of the unlabeled data.

## 4.5 Practical Implementation

Our algorithm consists of two main steps. In the first step, we construct a network, $\vartheta_\phi$, where $\phi$ are learnable parameters. We then use samples from $\mathcal{D}^{\text{UN}}$ and $\mathcal{D}^{\text{MIX}}$ to estimate the objective function $g(\vartheta)$. In the second step, after obtaining $\phi$ from the first step, we calculate the occupancy correction $\tau^*$ and use it to update the Q and V functions as described in Section 4.3. For the Q-updates, we utilize the following empirical objective: $\widetilde{F}(Q|V) = \sum_{(s,a,s') \sim \mathcal{D}^{\text{MIX}}}[\tau^*(s,a)r^Q(s,a,s') + \psi(r^Q(s,a,s'))] - (1-\gamma)\sum_{s^0 \sim \mathcal{D}^{\text{MIX}}} V(s)$, where the reward function is computed as $r^Q(s,a,s') = Q(s,a) - \gamma V^Q(s')$. We choose the $\chi^2$-divergence for the reward regularizer $\psi(t) = 0.1t^2 - t$, a popular choice in inverse Q-learning algorithm. For policy extraction we utilize the following practical weighted behavior cloning (WBC) with the objective: $\max_\pi \left\{ \sum_{(s,a) \sim \mu} \exp(A(s,a)) \log \pi(a|s) \right\}$, where $A(s,a)$ is the advantage function defined as $A(s,a) = (Q(s,a) - V(s))/\beta$ [37, 35].

## 5 Experiments

We evaluate our algorithm in the context of unconstrained RL and safe RL (i.e., constrained RL) using the Mujoco [42] and Safety-Gym [38, 21] domains, respectively. In the unconstrained task, we aim to test whether our algorithm can identify undesirable behaviors (from random or low-reward demonstrations) and avoid them. In the safety learning task, we focus on assessing whether our algorithm can satisfy safety constraints while maintaining reasonable rewards. Safe RL is also the main focus of SafeDICE in their experiments.

### 5.1 Experiment Setting

**Baselines.** We compare our algorithm against several baseline methods: (i) **BC**, which learns from the entire unlabeled dataset (BC-mix); (ii) **IPL** [17], a state-of-the-art algorithm in preference based RL; (iii) **DWBC** [45] and (iv) **LS-IQ** [3], two leading IL algorithms for learning from expert demonstrations, adapted to avoid undesirable demonstrations; and (v) **SafeDICE** [20], specifically designed for learning from undesirable demonstrations. We also include comparisons with modified baselines (similar to DWBC); however, due to space constraints, these are presented in Appendix E.7. Detailed implementation of these baselines is provided in Appendix D.3.

**Environments and Dataset Generation.** For our experiments, we define the components of the undesirable dataset $\mathcal{D}^{\text{UN}}$ and the unlabeled dataset $\mathcal{D}^{\text{MIX}}$. In the Mujoco experiments, we use the official D4RL dataset [10], which consists of three performance levels: random, medium, and expert. We combine the random and medium levels to create the undesirable dataset $\mathcal{D}^{\text{UN}}$ and use all three performance levels to construct the unlabeled dataset $\mathcal{D}^{\text{MIX}}$. In the safety learning experiments, we train two policies: a constrained policy (using PPO-Lagrangian [38] to achieve low cost performance) and an unconstrained policy (using PPO [41] maximizes the return while ignoring the cost signals). Data is collected from both policies, with the unconstrained policy is used to generate high-cost

demonstrations for undesirable dataset $\mathcal{D}^{\mathrm{UN}}$, and both datasets are combined to form the unlabeled dataset $\mathcal{D}^{\mathrm{MIX}}$. More details about the datasets can be found in Appendix D.2.

**Evaluation Metrics.** For the Mujoco domain, a higher return indicates better performance. In the Safety-Gym domain, the ideal outcome is to achieve the lowest cost possible while not significantly sacrificing the return. All experiments are run with at least 5 training seeds.

**Experiment Concerns.** Throughout the experiments, we aim to address several key questions: (**Q1**) *How does UNIQ perform compared to other baselines in the context of unconstrained and safe RL?* (**Q2**) *How does the presence of an undesirable dataset contribute to the performance of UNIQ and other baselines?* (**Q3**) *How does the quality of unlabeled dataset affect the performance of all the algorithms?* (**Q4**) *What happens if we do not use the unlabeled dataset?* (**Q5**) *We use WBC for the policy extraction; what if we directly extract the policy from the Q-function?* In the main paper, we provide experiments for (**Q1**), (**Q2**), and (**Q3**), while the other questions and some additional experiments are addressed in the appendix.

## 5.2 Main Comparison

| | | BC-mix | IPL | LS-IQ | DWBC | SafeDICE | UNIQ | Expert |
|---|---|---|---|---|---|---|---|---|
| Halfcheetah | Return ↑ | $17.9_{\pm6.3}$ | $3.2 \pm 0.7$ | $-2.8_{\pm0.3}$ | $2.2_{\pm0.0}$ | $3.1_{\pm0.9}$ | $\mathbf{75.7}_{\pm6.8}$ | $84.6_{\pm7.5}$ |
| Ant | Return ↑ | $86.3_{\pm11.8}$ | $84.5_{\pm13.1}$ | $1.8_{\pm5.6}$ | $84.6_{\pm12.4}$ | $4.4_{\pm2.6}$ | $\mathbf{104.4}_{\pm10.5}$ | $117.5_{\pm11.0}$ |
| Hopper | Return ↑ | $3.7_{\pm2.9}$ | $11.3_{\pm14.3}$ | $1.0_{\pm0.3}$ | $14.8_{\pm15.7}$ | $51.0_{\pm3.1}$ | $\mathbf{73.5}_{\pm20.6}$ | $110.8_{\pm1.2}$ |
| Walker2d | Return ↑ | $10.7_{\pm8.7}$ | $11.8_{\pm24.2}$ | $-0.1_{\pm0.1}$ | $38.8_{\pm24.7}$ | $41.7_{\pm4.1}$ | $\mathbf{105.9}_{\pm4.0}$ | $107.9_{\pm0.9}$ |
| Point-Goal | Return ↑ | $27.1_{\pm0.1}$ | $26.9_{\pm0.1}$ | $-6.8 \pm 4.4$ | $26.9_{\pm0.1}$ | $27.0_{\pm0.1}$ | $23.4_{\pm0.4}$ | $25.9_{\pm0.2}$ |
| | Cost ↓ | $48.8_{\pm2.9}$ | $52.7_{\pm3.4}$ | $18.0 \pm 29.3$ | $45.8_{\pm3.4}$ | $46.8_{\pm3.1}$ | $\mathbf{27.1}_{\pm3.0}$ | $26.0_{\pm2.6}$ |
| Car-Goal | Return ↑ | $34.1_{\pm0.5}$ | $34.7_{\pm0.3}$ | $-0.6 \pm 2.5$ | $32.8_{\pm0.7}$ | $33.5_{\pm0.7}$ | $27.9_{\pm0.8}$ | $26.2_{\pm0.7}$ |
| | Cost ↓ | $52.0_{\pm4.2}$ | $54.4_{\pm3.7}$ | $58.5 \pm 41.7$ | $47.4_{\pm3.8}$ | $50.5_{\pm4.0}$ | $\mathbf{31.0}_{\pm2.8}$ | $23.6_{\pm2.8}$ |
| Point-Button | Return ↑ | $17.6_{\pm0.7}$ | $16.9_{\pm0.9}$ | $-13.3 \pm 6.7$ | $17.2_{\pm0.9}$ | $15.1_{\pm0.5}$ | $12.6_{\pm1.4}$ | $14.4_{\pm1.0}$ |
| | Cost ↓ | $120.2_{\pm10.0}$ | $124.8_{\pm11.3}$ | $11.0 \pm 10.2$ | $123.5_{\pm14.4}$ | $91.0_{\pm6.4}$ | $\mathbf{23.0}_{\pm4.7}$ | $30.6_{\pm3.3}$ |
| Car-Button | Return ↑ | $17.6_{\pm0.7}$ | $17.2_{\pm0.8}$ | $-8.6 \pm 4.8$ | $17.1_{\pm1.0}$ | $17.4_{\pm0.6}$ | $12.7_{\pm1.1}$ | $14.0_{\pm0.8}$ |
| | Cost ↓ | $241.6_{\pm15.3}$ | $257_{\pm12.6}$ | $28.0 \pm 27.0$ | $249.2_{\pm20.9}$ | $201.3_{\pm10.8}$ | $\mathbf{148.6}_{\pm18.7}$ | $107.6_{\pm8.9}$ |

Table 1: Comparison results for Mujoco (i.e. unconstrained RL) and Safety-gym (i.e. constrained RL) tasks.

In this section, we aim to answer question (**Q1**), comparing UNIQ with the mentioned baselines across four Mujoco tasks and four Safety-Gym tasks. For the Mujoco tasks, we use 5 trajectories each from the random and medium datasets to construct the undesirable dataset $\mathcal{D}^{\mathrm{UN}}$, and 500, 500, and 100 trajectories from the random, medium, and expert datasets, respectively, to construct the unlabeled dataset $\mathcal{D}^{\mathrm{MIX}}$. For the Safety-Gym tasks, we use 100 high-cost trajectories for the undesirable dataset $\mathcal{D}^{\mathrm{UN}}$ and combine 1600 high-cost and 400 low-cost trajectories for the unlabeled dataset $\mathcal{D}^{\mathrm{MIX}}$. To facilitate comparison, we provide the performance of policies learned from desirable data (the expert dataset in Mujoco and the constrained dataset in Safety-Gym), which serve as the baseline for the highest performance. We note that our experiments require more samples than standard learning-from-expert experiments. This is intuitive, as *learning from mistakes* typically requires more examples than learning from good demonstrations.

The experimental results are shown in Table 1. Overall, BC-mix, LS-IQ, and IPL fail to meet our objectives (except in the Ant environment, where BC using all datasets seems to achieve good performance). These experiment results also show that the state-of-the-art algorithm in imitation learning, LS-IQ, does not perform well in the "avoid bad" scenario. DWBC outperforms BC-mix in Mujoco by using a discriminator to understand the undesirable dataset but struggles in the Safety-Gym tasks. Meanwhile, SafeDICE slightly outperforms DWBC in the Safety-Gym context. In general, UNIQ is the highest-performing algorithm in the Mujoco tasks and achieves closest (both in terms of return and costs) performance to policies learned from only the expert demonstrations (reported in "Expert" Column).

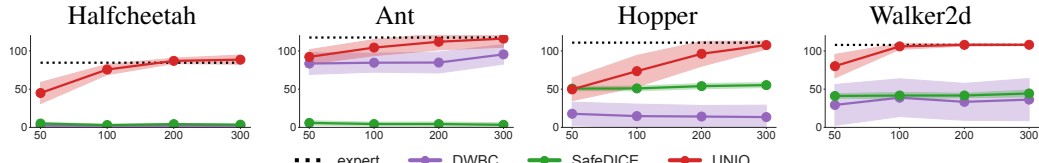

Figure 3: Comparison of results (MuJoCo tasks) for different sizes of expert demonstrations included in the unlabeled dataset, shaded by standard deviation.

## 5.3 Ablation Studies

**Impact of the Quality of the Unlabeled Dataset.** To address **Question (Q3)**, we modify the quality of the unlabeled dataset $\mathcal{D}^{\text{MIX}}$ by adjusting the number of desirable demonstrations while keeping the same $\mathcal{D}^{\text{UN}}$ as Section 5.2. The detailed results are shown in Figure 3. Overall, **UNIQ** consistently achieves the highest performance and adapts effectively to variations in dataset quality. The corresponding training curves for Mujoco tasks and Safety-gym tasks can be found in Appendix E.1.

**Impact of the Size of the Undesirable Dataset.**
We address **(Q2):** *the impact of the undesirable dataset on the final performance*. Detailed results are presented in Figure 4, where the $x$-axis represents the number of undesirable trajectories in $\mathcal{D}^{\text{UN}}$ and keeping the same $\mathcal{D}^{\text{MIX}}$ as Section 5.2. Overall, the size of the dataset significantly influences the final cost across all algorithms. As the size of the undesirable dataset increases, UNIQ achieves the lowest cost, albeit with a trade-off in return. Detailed learning curves for Mujoco and Safety-Gym can be found in the Appendix E.3. Furthermore, we discuss methods to control the conservativeness (trade-off between return and cost) of our approach in the Appendix E.10.

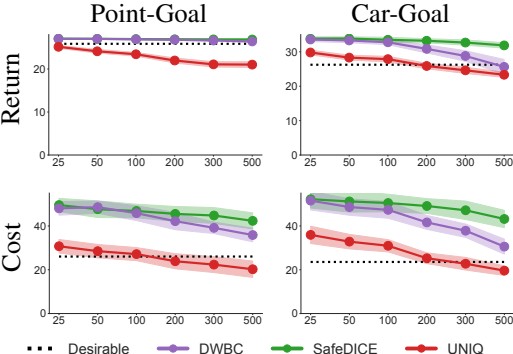

Figure 4: Comparison results for different sizes of the undesirable dataset in Safety gym, shaded by standard deviation.

## 6 Conclusion, Future Work

**Conclusion.** We have developed **UNIQ**, a principled framework designed to facilitate learning from undesirable demonstrations. UNIQ formulates the learning process as updates on $Q$- and $V$-functions based on convex loss functions, enabling unbiased training using unlabeled datasets of unknown quality. Additionally, UNIQ requires minimal hyper-parameter tuning, as it does not introduce any additional hyperparameters beyond those typically used in inverse Q-learning algorithms. Furthermore, UNIQ demonstrates superior performance in generating safe policies across several safe reinforcement learning experiments, outperforming other baseline methods.

**Limitations and Future work.** There are some aspects that have not been addressed in this paper, as they are too significant to be fully explored here. For instance, we assume the presence of only one set of undesirable demonstrations, whereas multiple datasets of varying quality could be leveraged to enhance the training. Additionally, each undesirable trajectory may not be undesirable in its entirety, as it could contain some good actions. Extracting the good parts from undesirable demonstrations could improve sample efficiency but introduces new challenges that warrant further investigation.

## Acknowledgments and Disclosure of Funding

This research is supported by the Singapore International Graduate Award (SINGA) Scholarship from the Agency for Science, Technology and Research, Singapore.

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

# Appendix

The appendix contains the following details:

**Missing Proofs:** Refer to Appendix B for proofs omitted from the main paper.

**Experimental Details:** We provide information on:

- Baseline implementation (Appendix D.3)
- Hyper-parameter selection for each task domain (Appendix D.4)
- Task descriptions (Appendix D.1)
- Generation of undesirable and unlabeled datasets (Appendix D.2)

**Additional Experiments:** We address the following remaining questions:

- (**Q2**) How does the presence of an undesirable dataset contribute to the performance of UNIQ and other baselines? (Appendix E.1)
- (**Q3**) How does the quality of unlabeled dataset affect the performance of all the algorithms? (Appendix E.2 and Appendix E.3)
- (**Q4**) What happens if we do not use the unlabeled dataset (the training is solely based on the undesirable dataset)? (Appendix E.5)
- (**Q5**) We use WBC for the policy extraction; what if we directly extract the policy from the Q-function? (Appendix E.4)

Additionally, we present experiments demonstrating the performance of UNIQ and other baseline methods on the dataset from the safeDICE paper (see Appendix E.12), performance in the Mujoco-velocity benchmark(see Appendix E.11), comparison results using CVaR costs (see Appendix E.9), How does $\tau$ effect to the overall results? (Appendix E.6), What if we want to avoid all possible undesirable performance in Safetygym (low-reward low-cost, low-reward high-cost, high-reward high-cost)? (Appendix E.8), and how to control the conservativeness in the Safety-gym tasks(see Appendix E.10).

# A  Pseudo Code of UNIQ

Below we present a pseudo code of our UNIQ algorithm.

---

**Algorithm 1 UNIQ**: **UN**desired Demonstrations driven **I**nverse **Q**-Learning

---

**Require:** $\mathcal{D}^{\text{UN}}, \mathcal{D}^{\text{MIX}}, \vartheta_\phi, \pi_\theta, N_\mu, N, Q_{w_q}$ and $V_{w_v}$ networks.
 1: # Estimating the occupancy correction $\tau^*$
 2: **for** certain number of iterations: $i = 1...N_\mu$ **do**
 3:     Update $(\phi)$ to maximize $g(\vartheta_\phi)$.
 4: **end for**
 5: # Train Q and V, and policy functions
 6: **for** certain number of iterations $i = 1...N$ **do**
 7:     Update $w_q$ to minimize $\widetilde{F}(Q_{w_q}|V_{w_v})$
 8:     Update $w_v$ to minimize the Extreme-V function: $J(V_{w_v}|Q_{w_q})$
 9:     Update $\theta$ via the WBC: $\max_\pi \left\{ \sum_{(s,a)\sim\mathcal{D}^{\text{MIX}}} \exp(A(s,a)) \log \pi(a|s) \right\}$
10: **end for**

---

# B  Missing Proofs

We provide proofs that are omitted in the main paper.

## B.1  Proof of Proposition 4.1

**Proposition.**  *The following statements hold:*

*(i) The function $L(\pi, Q)$ is convex in $\pi$ and the problem $\min_\pi L(\pi, Q)$ has a unique optimal solution at $\pi^Q(s,a) = \frac{\exp(Q(s,a)/\beta)}{\sum_{a'}\exp(Q(s,a')/\beta)}$.*

*(ii) The learning objective function can be simplified as:*

$$\min_Q \min_\pi \{L(\pi, Q)\} = \min_Q \left\{ \mathcal{F}(Q) = \mathbb{E}_{\rho^{\text{UN}}}[r^Q(s,a)] - (1-\gamma)\mathbb{E}_{s_0}[V^Q(s_0)] + \psi(r^Q) \right\}$$

*where $V^Q(s) = \beta \log \left( \sum_a \mu(a|s) \exp(Q(s,a)/\beta) \right)$ and $r^Q(s,a) = Q(s,a) - \gamma \mathbb{E}_{s'\sim P(\cdot|s,a)} V^Q(s')$.*

*Proof.*  We first express the second and third terms of the objective function $L(\pi, Q)$ in 5 as:

$$\mathbb{E}_{\rho_\pi}[\mathcal{T}^\pi[Q](s,a)] + H(\pi) = \mathbb{E}_{\rho_\pi}[Q(s,a) - \gamma\mathbb{E}_{s'}[V^\pi(s')]] - \beta\mathbb{E}_{\rho^\pi}[\log \frac{\pi(s,a)}{\mu(a|s)}]$$

$$= \mathbb{E}_{\rho_\pi}[Q(s,a) - \beta \log \frac{\pi(s,a)}{\mu(a|s)} - \gamma\mathbb{E}_{s'}[V^\pi(s')]] = \mathbb{E}_{\rho_\pi}[V(s) - \gamma\mathbb{E}_{s'\sim P(\cdot|s,a)}[V^\pi(s')]]$$

$$= (1-\gamma)\mathbb{E}_{s_0\sim P_0}[V^\pi(s_0)].$$

Thus, the objective function becomes:

$$L(\pi, Q) = \mathbb{E}_{\rho^{\text{UN}}}[Q(s,a) - \gamma\mathbb{E}_{s'}[V^\pi(s')]] - (1-\gamma)\mathbb{E}_{s_0\sim P_0}[V^\pi(s_0)] + \sum_{s,a} \psi(Q(s,a) - \gamma\mathbb{E}_{s'}[V^\pi(s')]).$$

We now observe that $V^\pi(s) = \mathbb{E}_{a\sim\pi(a|s)}[Q(s,a) - \beta\log\frac{\pi(a|s)}{\mu(a|s)}]$ is concave in $\pi$. Therefore, both terms $\mathbb{E}_{\rho^{\text{UN}}}[Q(s,a) - \gamma\mathbb{E}_{s'}[V^\pi(s')]]$ and $-(1-\gamma)\mathbb{E}_{s_0\sim P_0}[V^\pi(s_0)]$ are convex in $\pi$. Additionally, since $\psi(t)$ is convex and non-increasing in $t$, and $Q(s,a) - \gamma\mathbb{E}_{s'}[V^\pi(s')]$ is convex in $\pi$, each function $\psi(Q(s,a) - \gamma\mathbb{E}_{s'}[V^\pi(s')])$ is convex in $\pi$. Thus, combining all terms, we conclude that $L(\pi, Q)$ is convex in $\pi$.

Furthermore, each term $Q(s,a) - \gamma\mathbb{E}_{s'}[V^\pi(s')]$, $-(1-\gamma)\mathbb{E}_{s_0\sim P_0}[V^\pi(s_0)]$, and $\psi(Q(s,a) - \gamma\mathbb{E}_{s'})$ strictly decreases in $V^\pi$, implying that the minimization of $L(\pi, Q)$ over $\pi$ is achieved when $V^\pi(s)$

is maximized for all $s$. Since $V^\pi(s)$ is strictly concave in $\pi$, maximizing $V^\pi(s)$ over $\pi$ has a unique optimal solution:

$$\pi^Q(a|s) = \frac{\exp(Q(s,a)/\beta)}{\sum_a \exp(Q(s,a)/\beta)}.$$

This validates part (i) of the theorem.

For part (ii), we observe that the problem $\max_\pi V^\pi(s)$ has the optimal solution $\pi^Q$ as shown above, and the optimal value is:

$$\max_\pi V^\pi(s) = \max_\pi \left\{ \sum_a \pi(a|s)Q(s,a) - \beta\pi(a|s)\log\frac{\pi(a|s)}{\mu(a|s)} \right\}$$

$$= \beta\log\left( \sum_a \mu(a|s)\exp(Q(s,a)/\beta) \right) \stackrel{\text{def}}{=} V^Q(s).$$

This directly leads to:

$$\min_Q \min_\pi \{L(\pi, Q)\} = \min_Q \left\{ \mathcal{F}(Q) = \mathbb{E}_{\rho^{\text{UN}}}[r^Q(s,a)] - (1-\gamma)\mathbb{E}_{s_0}[V^Q(s_0)] + \psi(r^Q) \right\},$$

as required.

$\square$

## B.2 Proof of Proposition 4.2

**Proposition:** *The learning objective function $F(Q)$ is not convex in $Q$, even without any regularizer.*

*Proof.* We rewrite the function $F(Q)$ (without the regularizer $\psi(\cdot)$):

$$F(Q) = \mathbb{E}_{\rho^{\text{UN}}}\left[ Q(s,a) - \gamma\mathbb{E}_{s'}[V^Q(s')] \right] - (1-\gamma)\mathbb{E}_{s_0}[V^Q(s_0)],$$

where $V^Q(s) = \beta\log\left(\sum_a \mu(a|s)e^{Q(s,a)/\beta}\right)$ is a log-sum-exp function, which is convex in $Q$. Since $V^Q(s)$ is convex in $Q$, all the components associated with $V^Q(s)$ in $F(Q)$ are concave in $Q$. This directly implies the non-convexity of $F(Q)$.

When the regularizer is included, it can be observed that if $\psi(\cdot)$ is convex in $Q$, then $\psi(r^Q(s,a))$ is also convex in $Q$. As a result, the objective function $F(Q)$ takes the form of a difference-of-convex program. $\square$

## B.3 Proof of Proposition 4.3

**Proposition:** *Under any convex regularizer $\psi$, both $\widetilde{F}(Q|V)$ and $J(V|Q)$ are convex in $Q$ and $V$, respectively.*

*Proof.* We first write the loss function for the $Q$-updates as:

$$\widetilde{F}(Q|V) = \mathbb{E}_{\rho^{\text{UN}}}\left[ Q(s,a) - \gamma\mathbb{E}_{s'}[V(s')] \right] - (1-\gamma)\mathbb{E}_{s_0}[V(s_0)] + \psi\big(Q(s,a) - \gamma\mathbb{E}_{s'}[V(s')]\big),$$

where:

- The first term is linear in $Q$,

- The second term is a constant (since the $V$-function is fixed when updating the $Q$-function), and

- The last term involves a convex function of $Q$, making it convex.

Thus, $\widetilde{F}(Q|V)$ is convex in $Q$.

For the convexity of $J(V|Q)$, we first recall its formulation:

$$J(V|Q) = \mathbb{E}_{(s,a)\sim\mu}\left[ e^{\frac{Q(s,a)-V(s)}{\beta}} - \left(\frac{Q(s,a)-V(s)}{\beta}\right) - 1 \right].$$

The convexity of $J(V|Q)$ can be observed as follows:

- The first term is an exponential function of $V$, which is convex in $V$,

- The remaining terms are either linear in $V$ or constants.

Therefore, $J(V|Q)$ is convex in $V$. $\qquad\square$

## C  Connection between UNIQ and Statistical Distance Maximization

In this section, we connect the new MaxEnt objective function of UNIQ with the concept of statistical distance. Specifically, we show that solving the new MaxEnt objective $\min_r \min_\pi L(\pi, r)$ in (4) is equivalent to maximizing a statistical distance between the occupancy measures of the learning policy and the undesirable policy. In particular, if the feasible set of the reward function is unrestricted, the following equality holds:

$$\min_r \min_\pi \{L(\pi, r)\} = -\max_\pi \{d_\psi(\rho^\pi, \rho^{\mathrm{UN}}) - H(\pi)\}, \tag{9}$$

where $d_\psi(\rho^\pi, \rho^{\mathrm{UN}}) = \psi^*(\rho^\pi - \rho^{\mathrm{UN}})$, and $\psi^*$ is the convex conjugate of the convex function $\psi$, i.e.,

$$\psi^*(t) = \sup_z (tz - \psi(z)).$$

To verify (9), we write the objective function as:

$$L(\pi, r) = \sum_{s,a} r(s, a) \left(\rho^{\mathrm{UN}}(s, a) - \rho^\pi(s, a)\right) + \psi(r) - H(\pi).$$

Thus, the minimization of $L(\pi, r)$ over $r$ can be expressed as:

$$\min_r L(\pi, r) = H(\pi) + \min_r \left\{\sum_{s,a} r(s, a) \left(\rho^{\mathrm{UN}}(s, a) - \rho^\pi(s, a)\right) + \psi(r)\right\}$$

$$= H(\pi) - \max_r \left\{\sum_{s,a} r(s, a) \left(\rho^\pi(s, a) - \rho^{\mathrm{UN}}(s, a)\right) - \psi(r)\right\}.$$

Since the feasible set of the reward function $r$ is unrestricted, we have:

$$\max_r \left\{\langle r, \rho^\pi - \rho^{\mathrm{UN}}\rangle - \psi(r)\right\} = \psi^*(\rho^\pi - \rho^{\mathrm{UN}}).$$

This allows us to rewrite the training problem as:

$$\min_r \min_\pi \{L(\pi, r)\} = \min_\pi \left\{H(\pi) - \psi^*(\rho^\pi - \rho^{\mathrm{UN}})\right\}$$

$$= -\max_\pi \left\{\psi^*(\rho^\pi - \rho^{\mathrm{UN}}) - H(\pi)\right\}$$

$$= -\max_\pi \left\{d_\psi(\rho^\pi, \rho^{\mathrm{UN}}) - H(\pi)\right\}.$$

We thus obtain the desired equation in (9), proving the equivalence.

## D  Experimental settings

### D.1  Environmental Details

#### D.1.1  Mujoco

The MuJoCo locomotion benchmark focuses on a set of environments designed to evaluate the ability of reinforcement learning algorithms to control agents with complex, multi-joint dynamics. These tasks involve training agents to move efficiently and stably in simulated environments. The illustrations are shown in Figure 5.

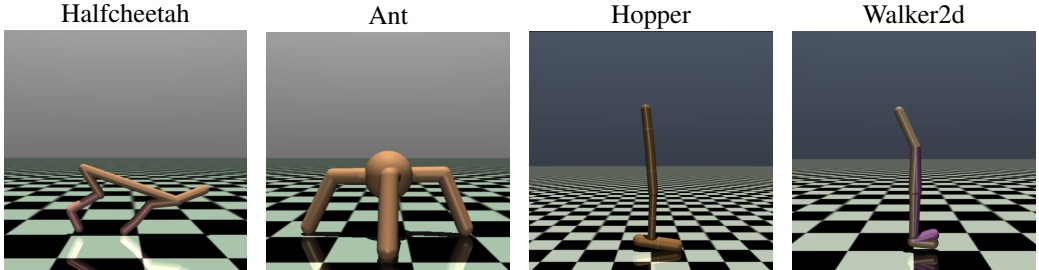

Figure 5: Mujoco environments.

### D.1.2 Safe-Gym

Safe-Gym is a collection of reinforcement learning environments designed with a focus on safety, built on top of the OpenAI Gym framework. It introduces constraints that simulate safety-critical scenarios commonly encountered in real-world applications. In Safe-Gym, agents are rewarded for completing task-specific objectives but face penalties for violating safety constraints, such as surpassing speed limits, colliding with obstacles, or entering restricted zones. These constraints enable Safe-Gym to replicate environments where safety is paramount, including robotic navigation in congested areas, autonomous vehicle control, and industrial automation. The environment features two types of agents: Point (an easy agent) and Car (a more challenging agent), as well as two types of tasks: Goal (easy) and Button (hard). Additionally, the environment dynamics change with each new episode, introducing variability and increasing the complexity of the tasks. Illustrations of these tasks are shown in Figure 6.

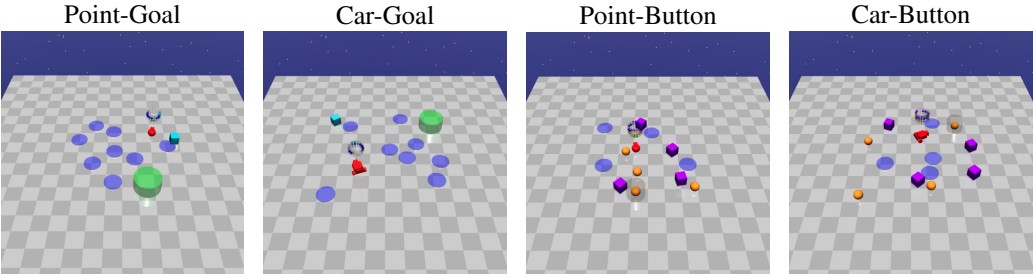

Figure 6: Safety-gym environments.

### D.1.3 Mujoco-velocity

From original Mujoco, Mujoco-Velocity is a specialized environment within the Mujoco physics simulation suite, focusing on controlling the velocity of two specific agents: Cheetah and Ant. These agents must complete locomotion tasks while adhering to safety constraints on their speed. The goal is to balance task performance with maintaining safe velocity limits. For instance, Cheetah must run as fast as possible while staying within predefined speed bounds to avoid penalties, mimicking real-world scenarios where exceeding speed limits can cause system failure or unsafe operations. Similarly, Ant must navigate through its environment without violating velocity constraints, ensuring stability and safety.

### D.2 Dataset generation details

In this paper, we tackle the problem of **Offline Learning with Undesirable Demonstrations** by utilizing two datasets:

- Unlabeled dataset $\mathcal{D}^{\text{MIX}}$: A large dataset comprising both desired and undesired demonstrations, reflecting real-world data (e.g., chat conversations, driving behaviors, treatment decisions, etc.).

- Undesired dataset $\mathcal{D}^{\mathrm{UN}}$: A smaller, accurate dataset containing demonstrations that exhibit behaviors we aim to avoid.

### D.2.1  Mujoco dataset

We use the offical D4RL dataset with three different performance: random, medium (sub-optimal), and expert. We then combine them to collect the dataset as follow:

- Unlabeled dataset $\mathcal{D}^{\mathrm{MIX}}$: We mix all three dataset into a large unlabeled dataset with a specific ratio.

- Undesired dataset $\mathcal{D}^{\mathrm{UN}}$: We mix random and medium dataset with the same ratio $(1:1)$.

### D.2.2  Safety-gym and Mujoco-velocity dataset

We simulate this scenario using the Safety-gym and Mujoco-velocity environments. First, we train both unconstrained and constrained policies using PPO [41] and PPO-lagrangian [38]. We then collect the training datasets as follows:

- Unlabeled dataset $\mathcal{D}^{\mathrm{MIX}}$: We roll out the constrained and unconstrained policies, mixing them at ratios of $(1:4)$.

- Undesired dataset $\mathcal{D}^{\mathrm{UN}}$: We roll out the unconstrained policy, gathering trajectories that violate the constraint.

The quality of Safety-gym dataset are shown in Table 2 while details information of Mujoco-velocity datasets are shown in Table 3.

|  | Point-Goal | Car-Goal | Point-Button | Car-Button |
|---|---|---|---|---|
| Mean Unconstrained return | $26.8 \pm 1.1$ | $35.2 \pm 2.1$ | $21.49 \pm 4.87$ | $17.82 \pm 5.57$ |
| Mean Constrained return | $25.3 \pm 2.1$ | $25.7 \pm 6.6$ | $13.71 \pm 6.44$ | $14.99 \pm 8.45$ |
| Mean Unconstrained cost | $57.8 \pm 38.9$ | $61.9 \pm 48.1$ | $138.9 \pm 77.3$ | $236.5 \pm 115.9$ |
| Mean Constrained cost | $22.3 \pm 28.0$ | $21.0 \pm 31.0$ | $27.4 \pm 34.1$ | $113.5 \pm 33.2$ |
| Cost Threshold | 25.0 | 25.0 | 25.0 | 100.0 |

Table 2: Safety-Gym expert policies performance.

|  | Cheetah | Ant |
|---|---|---|
| Mean Unconstrained return | $3027.4 \pm 400.6$ | $2972.2 \pm 1020.0$ |
| Mean Constrained return | $2751.2 \pm 11.6$ | $2830.0 \pm 145.5$ |
| Mean Unconstrained cost | $626.7 \pm 95.0$ | $624.0 \pm 231.0$ |
| Mean Constrained cost | $14.1 \pm 4.5$ | $18.7 \pm 4.4$ |
| Cost Threshold | 25.0 | 25.0 |

Table 3: Mujoco-velocity expert policies performance.

### D.3  Baseline implementation details

### D.3.1  BC

We use the orginal BC objective:

$$\min_{\pi} -\mathbb{E}_{s,a \sim \mathcal{D}} \log \pi(a|s) \tag{10}$$

### D.3.2 LS-IQ

We use the official implementation of LS-IQ in this link. We modify the objective to fit with our scenario when undesirable dataset available:

$$\min_Q \ \mathbb{E}_{s,a\sim\mathcal{D}^{\text{UN}}}\left[(Q(s,a)-Q_{min})^2\right] + \mathbb{E}_{s,a,s'\sim\mathcal{D}^{\text{MIX}}}\left[(Q(s,a)-r_{max}-\gamma V(s'))^2\right],$$

where $V(s) = Q(s,\pi(a|s)) + H(\pi)$; $Q_{min} = r_{min}/(1-\gamma)$; and $r_{min}$ and $r_{max}$ are predefined.

### D.3.3 IPL

We use the official implementation of IPL [17] from this link. The only difference between our setting and IPL is the pairwise dataset. We create pairwise comparisons from the unlabeled dataset and the undesired dataset and train the Q-function with the following new loss function:

$$P_{Q^\pi}[\sigma^{MIX} > \sigma^{UN}] = \frac{\exp\sum_t(\mathcal{T}^\pi Q)(s_t^{MIX},a_t^{MIX})}{\exp\sum_t(\mathcal{T}^\pi Q)(s_t^{MIX},a_t^{MIX}) + \exp\sum_t(\mathcal{T}^\pi Q)(s_t^{UN},a_t^{UN})},$$

where:

$$(\mathcal{T}^\pi Q)(s,a) = Q(s,a) - \gamma\mathbb{E}_{s'}[V^\pi(s')].$$

### D.3.4 DWBC

We use the official implementation of DWBC [45] from this link. We modify the algorithm to train a discriminator that assigns 0 to the undesired dataset $\mathcal{D}^{\text{UN}}$ and 1 to the unlabeled dataset $\mathcal{D}^{\text{MIX}}$ while keeping the Positive Unlabeled learning technique from the paper:

$$\min_\theta \ \eta\mathbb{E}_{(s,a)\sim\mathcal{D}^{\text{MIX}}}\left[-\log d_\theta(s,a,\log\pi)\right]$$
$$+ \mathbb{E}_{(s,a)\sim\mathcal{D}^{\text{UN}}}\left[-\log\left(1-d_\theta(s,a,\log\pi)\right)\right]$$
$$- \eta\mathbb{E}_{(s,a)\sim\mathcal{D}^{\text{MIX}}}\left[-\log\left(1-d_\theta(s,a,\log\pi)\right)\right].$$

### D.3.5 SafeDICE

We use the official implementation of SafeDICE [20] from this link. As the algorithm has been designed to solve this problem, we do not make any further modifications.

### D.4 Hyper-parameter selection

For fair comparison, we keep the same basic hyper-parameters across all the baselines which are detailed as follow for Mujoco, Safety-gym and Mujoco-velocity tasks as Table 4:

| HYPER PARAMETER | MUJOCO | SAFETY-GYM | MUJOCO-VELOCITY |
|---|---|---|---|
| ACTOR NETWORK | 256x3 | 256x3 | 256x3 |
| CRITIC NETWORK | 256x3 | 256x3 | 256x3 |
| VALUE NETWORK | 256x3 | 256x3 | 256x3 |
| TRAINING STEP | 1,000,000 | 1,000,000 | 1,000,000 |
| GAMMA | 0.99 | 0.99 | 0.99 |
| LR ACTOR | 0.0003 | 0.0001 | 0.0001 |
| LR CRITIC | {0.0001,0.0003} | 0.0003 | 0.0003 |
| LR VALUE | 0.0003 | 0.0003 | 0.0003 |
| LR DISCRIMINATOR | 0.0001 | 0.0001 | 0.0001 |
| BATCH SIZE | 256 | 256 | 256 |
| SOFT UPDATE CRITIC FACTOR | [0.001,0.01] | 0.005 | 0.005 |

Table 4: Hyper parameters.

In UNIQ, the $\alpha$ and $\beta$ parameters are selected in range $[0.5, 10.0]$. Lastly, we apply state normalization for Mujoco and Mujoco-velocity datasets as follow:

$$s_{\text{normalized}} = \frac{s - \mu}{\sigma}$$

Where:

$$\mu = \frac{1}{|\mathcal{D}|} \sum_{s' \in \mathcal{D}} s'$$

$$\sigma = \sqrt{\frac{1}{|\mathcal{D}|} \sum_{s' \in \mathcal{D}} (s' - \mu)^2}$$

## D.5 Computational Resource

Our experiments were carried out on a GPU cluster equipped with 8 NVIDIA RTX 3090 GPUs. Each experimental setup was run with five distinct training seeds in parallel, sharing a single GPU, eight CPU cores, and 64 GB of RAM. Within this shared configuration, completing one million training steps across all five seeds took approximately 30 minutes. The software environment was built using JAX version 0.4.28, with support for CUDA 12—specifically, CUDA version 12.3.2 and cuDNN version 8.9.7.29.

It is important to note that in the Safety-Gym domain, evaluation is relatively slow due to the use of 50 different environment random seeds for each (we have 100 evaluations in total). While the training phase takes around 30 minutes, the extensive evaluation increases the total runtime per experiment to approximately 2 hours.

# E  Additional Experiments

## E.1  Performance with Different Sizes of undesirable Dataset

In this section, we provide a complete experiment to evaluate the contribution of the undesirable dataset to our algorithm. We test our method with an increasing size of the undesirable dataset, $\mathcal{D}^{UN} = \{5, 10, 20, 50, 100, 200\}$ for the Mujoco tasks and $\mathcal{D}^{UN} = \{25, 50, 100, 200, 300, 500\}$. The full experiment results are shown in Figure 7 (Mujoco) and Figure 8 (Safety-Gym).

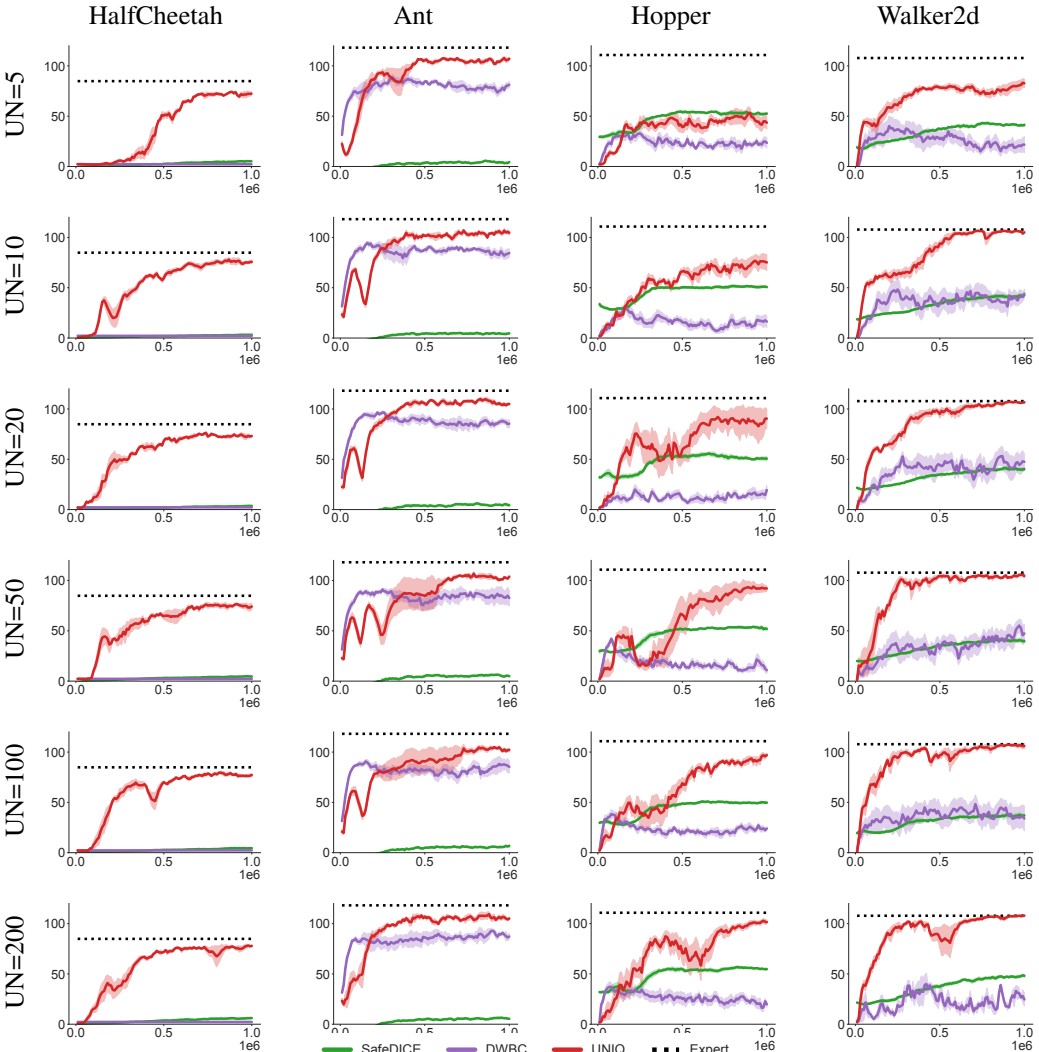

Figure 7: Comparison with different size of Undesirable dataset in Mujoco environments.

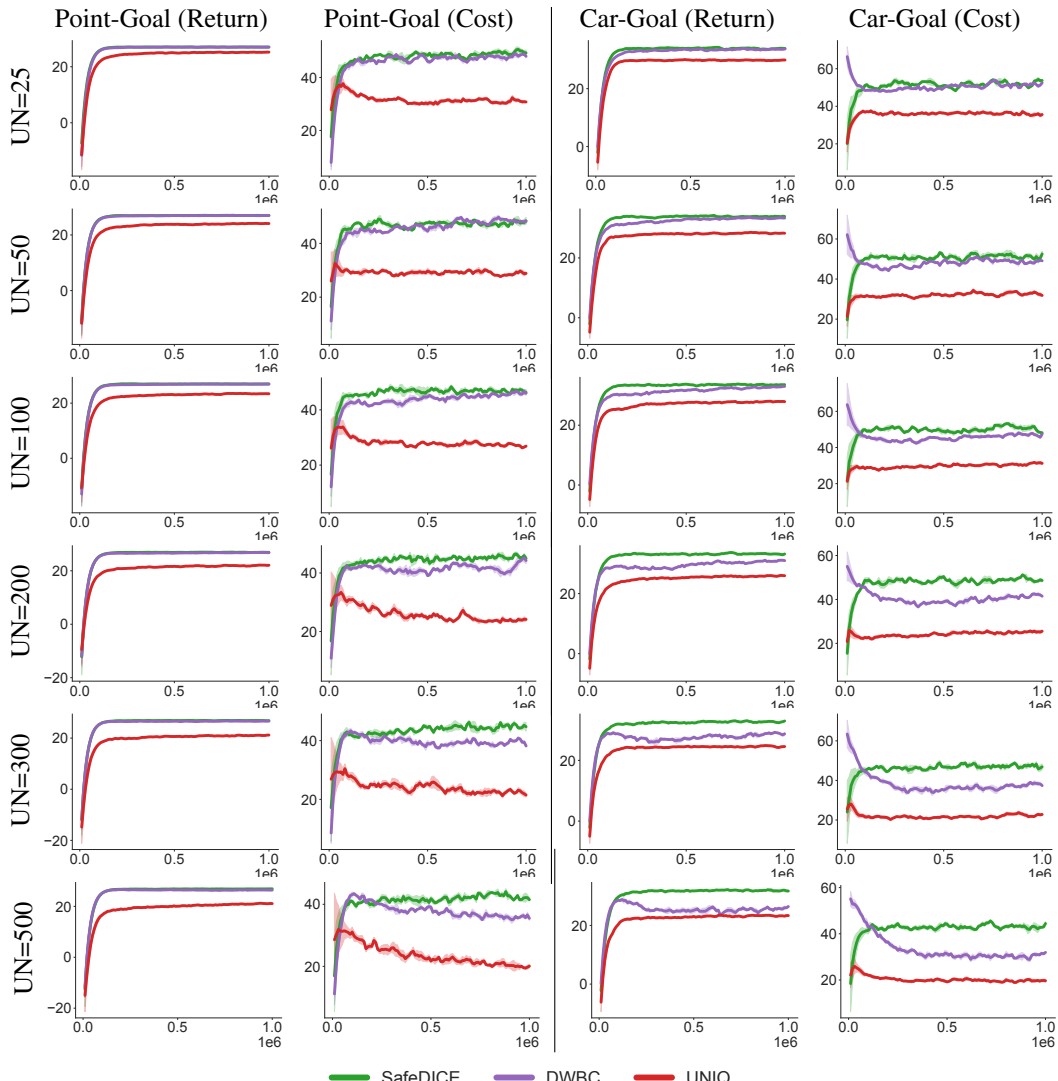

Figure 8: Comparison with different size of Undesirable dataset in Safety-gym environments.

## E.2 Performance with Different Number of Undesirable Demonstrations in the Unlabeled Dataset

We further change the number of undesirable demonstrations in the unlabeled dataset $\mathcal{D}^{\mathrm{MIX}}$ to examine how it impacts the performance of the algorithm. The experimental results are shown in Figure 9. The results show that when the quality of the dataset decreases significantly, the performance of all algorithms worsens. However, UNIQ still achieves the highest performance across all environments.

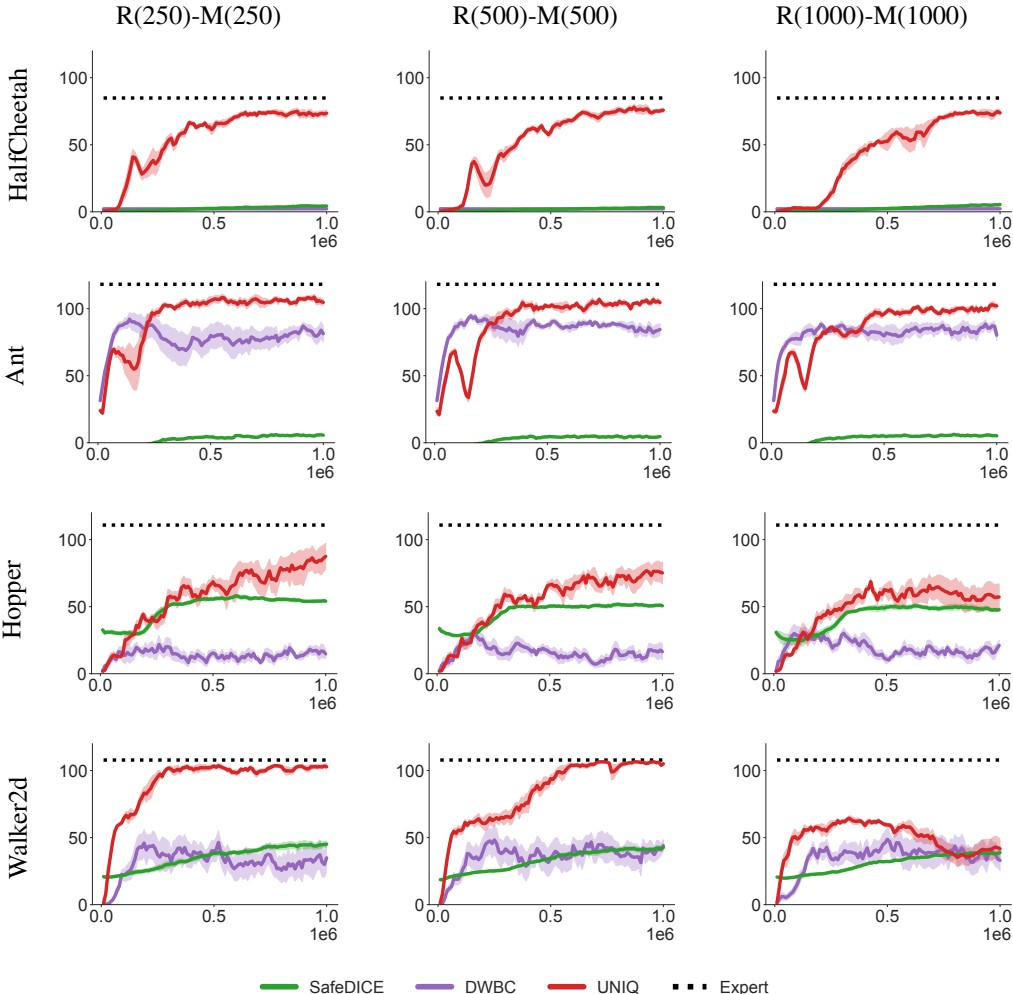

Figure 9: Comparison with different size of Undesirable dataset in Mujoco environments.

### E.3 Performance with Different Number of Desirable Demonstrations in the Unlabeled Dataset

We also test how changing the amount of desirable data in the unlabeled dataset affects the performance. The experimental results are shown in Figure 10 (Mujoco) and Figure 11 (Safety-Gym). We observe an increasing trend in performance as the size of the desirable data increases (higher returns in Mujoco and lower costs in Safety-Gym).

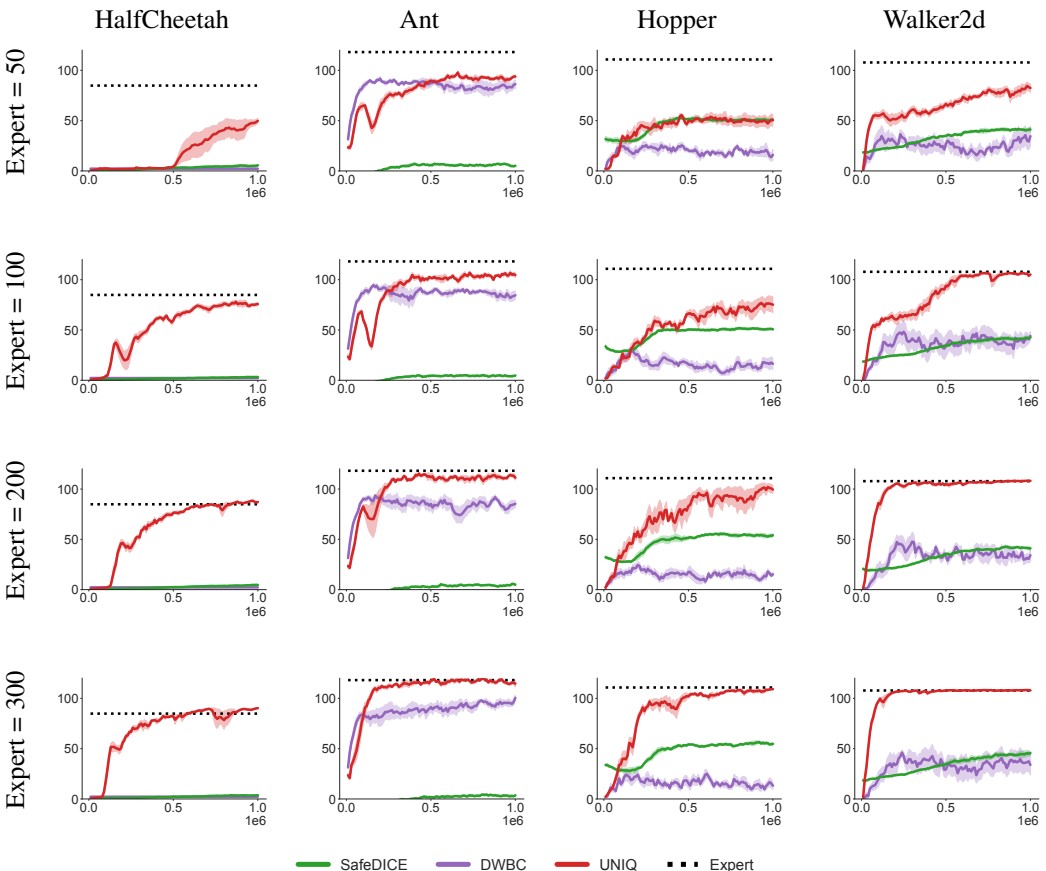

Figure 10: Comparison with different size of Undesirable dataset in Mujoco environments.

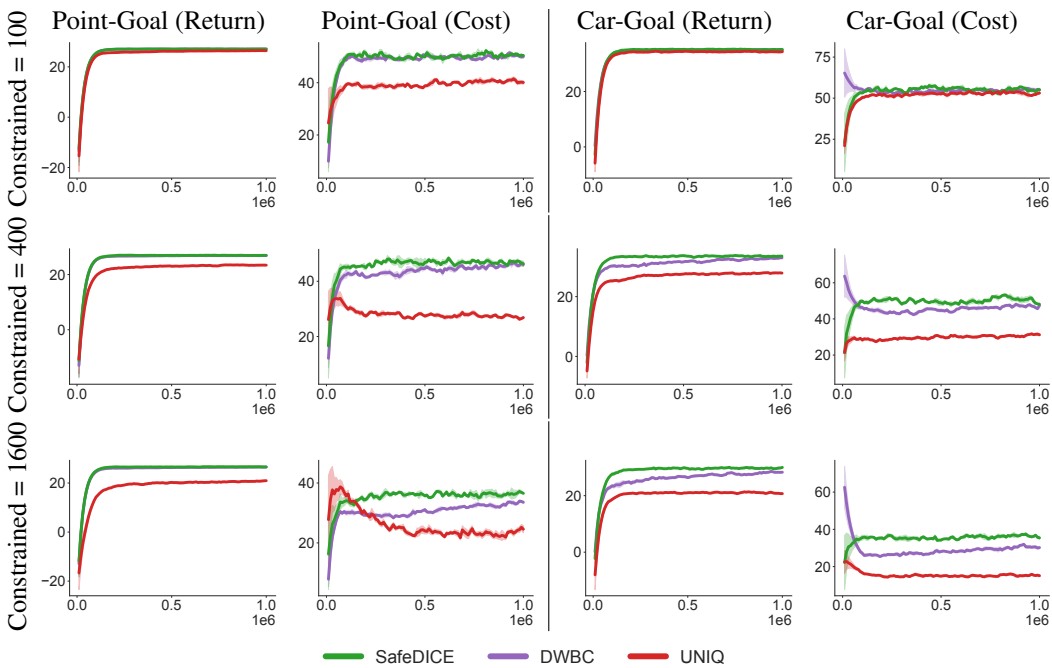

Figure 11: Comparison with different size of Undesirable dataset in Safety-gym environments.

## E.4 Direct Policy Extraction from Q-functions

As our actor update objective uses Weighted Behavioral Cloning, a question arises: what if we directly extract the actor from the Q-function [12, 3]? We conduct experiments to evaluate the performance of UNIQ when the actor is learned directly from the Q-function. The results are presented in Figure 12 and Figure 13..

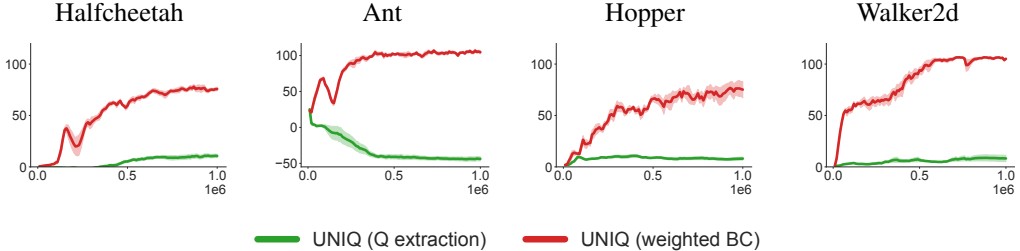

Figure 12: compare UNIQ BC update with Q extraction update in D4RL

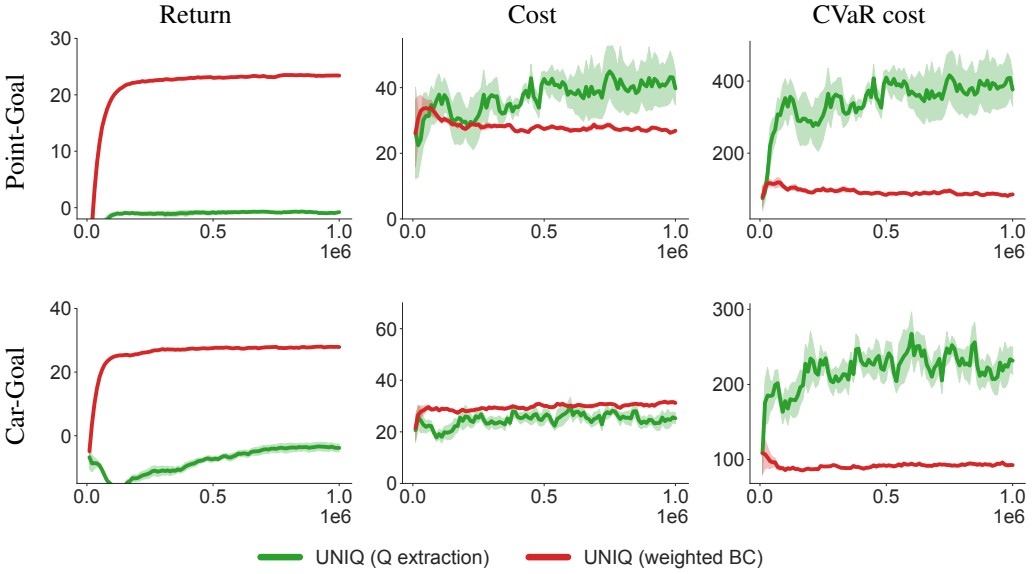

Figure 13: compare UNIQ BC update with Q extraction update in Safetygym

## E.5 BC without Unlabeled Dataset

As UNIQ uses an unlabeled dataset to learn by avoiding undesirable behaviors and learning from the remaining parts of the dataset, an interesting question arises: what if we only avoid the undesirable dataset (no unlabeled dataset is given)? To explore this, we train a Behavioral Cloning (BC) agent that, instead of following the dataset, explicitly avoids the actions from the undesirable dataset by minimizing the probability of those assigned actions, denoted as BC-UN. The detailed results are shown in Figure 14 (Mujoco) and Figure 15 (Safety-Gym).

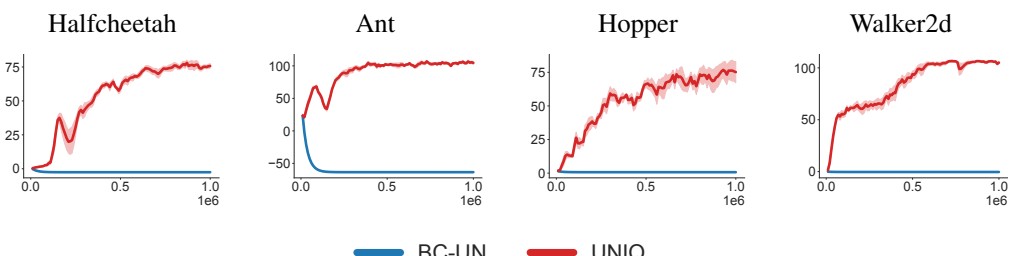

Figure 14: BC only avoid undesirable data in D4RL

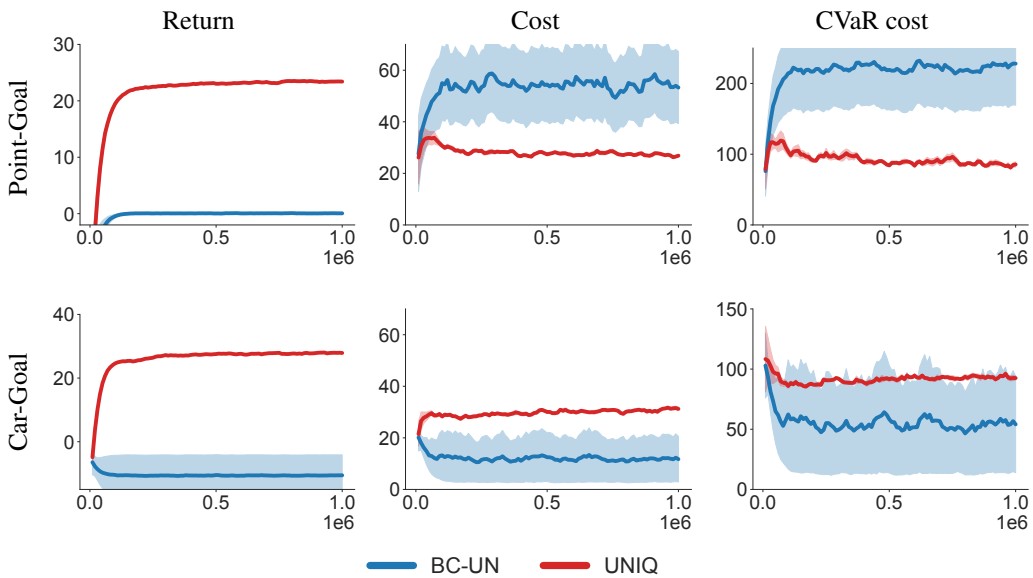

Figure 15: BC only avoid undesirable data in Safetygym

## E.6 ablation study with different $\tau$

| Task | $\tau = 1$ | Non-converged $\tau$ | Converged $\tau$ |
|------|-----------|---------------------|------------------|
| HalfCheetah | $18.5 \pm 5.6$ | $64.9 \pm 5.2$ | $\mathbf{75.7 \pm 6.8}$ |
| Ant | $88.5 \pm 8.2$ | $95.2 \pm 12.4$ | $\mathbf{104.4 \pm 10.5}$ |
| Hopper | $57.4 \pm 12.4$ | $58.0 \pm 14.6$ | $\mathbf{73.5 \pm 20.6}$ |
| Walker2d | $12.8 \pm 7.7$ | $72.0 \pm 6.1$ | $\mathbf{105.9 \pm 4.0}$ |

Table 5: Ablation study with different $\tau$: We maintain the same dataset construction as in the main experiment.

To assess the effect of occupancy ratio estimation on our algorithm, we conducted additional ablation studies using three configurations: (i) a uniform occupancy ratio ($\tau = 1$), (ii) a non-converged estimation of $\tau$, and (iii) a properly converged $\tau$. The results of this ablation study (for the unconstrained RL setting) are shown in Table 5.

## E.7 Additional comparison with modified baselines for avoiding undesirable

| Task | BC-remove-bad | IS-WBC | ILID | UNIQ |
|------|---------------|--------|------|------|
| HalfCheetah | $2.3 \pm 0.0$ | $2.2 \pm 0.0$ | $2.1 \pm 0.0$ | $\mathbf{75.7 \pm 6.8}$ |
| Ant | $36.2 \pm 19.6$ | $73.4 \pm 12.3$ | $\mathbf{106.44 \pm 4.3}$ | $104.4 \pm 10.5$ |
| Hopper | $19.7 \pm 18.9$ | $7.6 \pm 5.0$ | $5.9 \pm 6.5$ | $\mathbf{73.5 \pm 20.6}$ |
| Walker2d | $37.8 \pm 33.3$ | $18.8 \pm 19.3$ | $69.6 \pm 13.0$ | $\mathbf{105.9 \pm 4.0}$ |

Table 6: Additional comparison with BC-remove-bad, IS-WBC, and ILID: while keeping the dataset construction consistent with the main experiment.

Since our method is designed to learn from undesirable demonstrations, we evaluate it alongside several baseline algorithms that were originally developed for expert demonstrations but have been adapted for our setting:

- **BC-remove-bad**: We first train a classifier to detect and remove undesirable demonstrations from the unlabeled dataset. Behavior Cloning (BC) is then applied to the filtered data to learn the policy.

- **ISW-BC**: We modify the original discriminator objective to better suit the learning-from-undesirable-demonstrations framework. Specifically, we train the discriminator to differ-

entiate between undesirable and unlabeled demonstrations, rather than between expert and non-expert data as in the original method. This adaptation helps guide the policy away from undesirable behaviors without requiring expert data.

- **ILID**: We adapt the original ILID approach by removing all states in the unlabeled dataset that resemble those in the undesirable dataset. However, the original ILID also includes a regularization component that prevents the learned policy from deviating significantly from a BC-trained expert policy. Since expert demonstrations are unavailable in our setting, this regularization is excluded in our adaptation.

The comparative performance results are reported in Table 6. Overall, UNIQ yell the best performance in Mujoco tasks.

### E.8 Performance with 4 category types (low-cost low-reward, low-cost high-reward, high-cost low-reward, high-cost high-reward) of dataset in Safety-gym

| | | DWBC | SafeDICE | UNIQ | Expert |
|---|---|---|---|---|---|
| Point-Goal | Return | $20.1 \pm 1.1$ | $19.9 \pm 0.4$ | $\mathbf{24.3 \pm 0.8}$ | $25.9 \pm 0.2$ |
| | Cost | $47.0 \pm 4.5$ | $41.7 \pm 5.0$ | $\mathbf{26.0 \pm 4.4}$ | $26.0 \pm 2.6$ |
| Car-Goal | Return | $20.9 \pm 1.5$ | $22.0 \pm 3.6$ | $\mathbf{22.5 \pm 2.2}$ | $26.2 \pm 0.7$ |
| | Cost | $30.3 \pm 3.3$ | $30.5 \pm 7.7$ | $\mathbf{19.9 \pm 2.8}$ | $23.6 \pm 2.8$ |

Table 7: Performance of 4 different category of performance types on Point-Goal and Car-Goal Tasks.

The objective of this experiment is to evaluate whether the algorithm can successfully avoid all types of undesirable demonstrations in the Safety-Gym environment. While our main experiments (Table 1) primarily assess safety performance with minimal return degradation, this experiment focuses on the algorithm's ability to comprehensively reject undesirable trajectories.

To achieve this, we construct a dataset based on four categories of Safety-Gym performance: high-return low-cost (H-L), high-return high-cost (H-H), low-return low-cost (L-L), and low-return high-cost (L-H). The dataset is divided into two subsets: the undesirable dataset $\mathcal{D}^{UN}$, which consists of 50 trajectories each from the H-H, L-L, and L-H categories (totaling 150 trajectories); and the unlabeled dataset $\mathcal{D}^{MIX}$, which contains 500 trajectories from each category, resulting in 2,000 trajectories in total.

The experimental results are reported in Table 7. may initially appear counterintuitive—SafeDICE and UNIQ exhibit reduced cost despite the more challenging data. However, this outcome is likely due to a trade-off inherent in the policy behavior: lower rewards are often correlated with lower costs. In particular, policies that prioritize minimizing cost may opt for conservative behaviors, such as remaining stationary, which avoids hazards but also leads to diminished task progress and, consequently, lower returns.

### E.9 Comparisons with CVaR $10\%$ Cost for Safety-gym Tasks

We also report the CVaR $10\%$ cost for Safety-gym tasks, supporting the result of the Table 1 with CVaR is the mean of $10\%$ highest in cost trajectories during the evaluation process. The full results are shown in Table 8.

|  |  | BC-mix | LS-IQ | IPL | DWBC | SafeDICE | UNIQ |
|---|---|---|---|---|---|---|---|
| Point-Goal | Return | 27.1±0.1 | $-6.8 \pm 4.4$ | 26.9±0.1 | 26.9±0.1 | 27.0±0.1 | 23.4±0.4 |
|  | Cost | 48.8±2.9 | $18.0 \pm 29.3$ | 52.7±3.4 | 45.8±3.4 | 46.8±3.1 | 27.1±3.0 |
|  | CVaR | 115.4±7.7 | $133.3 \pm 240.7$ | 117.9±8.2 | 110.4±7.8 | 111.0±7.6 | **85.9± 18.9** |
| Car-Goal | Return | 34.1±0.5 | $-0.6 \pm 2.5$ | 34.7±0.3 | 32.8±0.7 | 33.5±0.7 | 27.9±0.8 |
|  | Cost | 52.0±4.2 | $58.5 \pm 41.7$ | 54.4±3.7 | 47.4±3.8 | 50.5±4.0 | 31.0±2.8 |
|  | CVaR | 132.8±10.6 | $311.1 \pm 208.9$ | 134.9±8.7 | 123.2±10.6 | 128.5±10.2 | **93.3 ± 8.4** |
| Point-Button | Return | $17.6 \pm 0.7$ | $-13.3 \pm 6.7$ | $16.9_{\pm 0.9}$ | $17.2 \pm 0.9$ | $15.1 \pm 0.5$ | $12.6 \pm 1.4$ |
|  | Cost | $120.2 \pm 10.0$ | $11.0 \pm 10.2$ | $124.8_{\pm 11.3}$ | $123.5 \pm 14.4$ | $91.0 \pm 6.4$ | $23.0 \pm 4.7$ |
|  | CVaR | $311.0 \pm 47.8$ | $58.2 \pm 89.1$ | $307.2 \pm 47.3$ | $309.6 \pm 63.7$ | $229.2 \pm 21.2$ | **$98.7 \pm 33.1$** |
| Car-Button | Return | $17.6 \pm 0.7$ | $-8.6 \pm 4.8$ | $17.2_{\pm 0.8}$ | $17.1 \pm 1.0$ | $17.4 \pm 0.6$ | $12.7 \pm 1.1$ |
|  | Cost | $241.6 \pm 15.3$ | $28.0 \pm 27.0$ | $257_{\pm 12.6}$ | $249.2 \pm 20.9$ | $201.3 \pm 10.8$ | $148.6 \pm 18.7$ |
|  | CVaR | $545.3 \pm 45.6$ | $195.9 \pm 204.1$ | $553.2 \pm 35.7$ | $550.5 \pm 66.3$ | **$410.5 \pm 34.3$** | $449.8 \pm 63.4$ |

Table 8: Full comparison results in Return, Cost, and CVaR $10\%$.

### E.10 Controlling Conservativeness in UNIQ

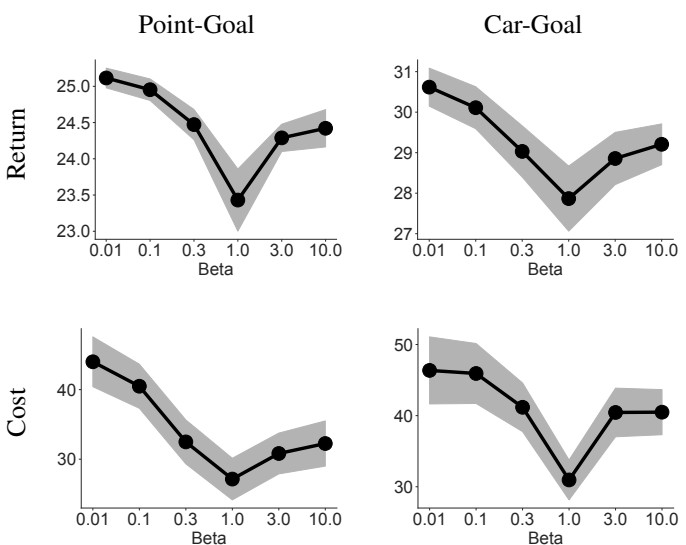

Figure 16: Comparison results of UNIQ with different $\alpha$ selections.

The main paper demonstrates that the policy returned by UNIQ achieves significantly lower costs (indicating safety) but, in some cases, also lower rewards compared to other imitation learning baselines. While this aligns well with our objective of learning safe policies by avoiding unsafe demonstrations, it also raises concerns about the algorithm's conservativeness.

In this section, we show that the conservativeness of UNIQ can be effectively controlled by introducing a parameter to the Weighted BC formulation. Specifically, we adjust the conservativeness of the algorithm by adding a parameter $\beta$ to the Weighted BC update:

$$\sum_{(s,a)\sim\mathcal{D}^{\text{Mix}}} \exp((Q_{w_q}(s,a) - V_{w_v}(s)) * \beta) \log \pi_\theta(a|s)$$

When $\beta = 1$, the Weighted BC theoretically returns the exact policy derived from Q-learning, as reported in the main paper. In contrast:

- As $\beta \to 0$, the Weighted BC returns a random policy.
- As $\beta \to \infty$, the resulting policy becomes deterministic, always selecting the best action with probability 1.

Thus, by varying $\beta$, we can deviate the outcome of the Weighted BC from the policy given by Q-learning, reducing the conservativeness of the learned policy.

To experimentally demonstrate this, we vary $\beta$ and report the corresponding returns and costs on four MuJoCo environments. The results are presented in Figure 16 and Table 9, showing how different values of $\beta$ impact the trade-off between safety and performance.

Figure 16 demonstrates that UNIQ achieves its safest (and most conservative) performance when $\beta = 1$. At this value, the policy prioritizes minimizing costs, making it the most risk-averse option. However, as $\beta$ deviates from 1, both the cost and return increase. This indicates that the Weighted BC formulation produces less conservative policies that are less safe but capable of achieving higher rewards.

Table 9 provides a more detailed breakdown of the costs and returns for different values of $\beta$. The results show that UNIQ can effectively balance safety and performance: by adjusting $\beta$, it is possible to achieve a safer policy (i.e., lower cost) while maintaining competitive returns (compared to other baselines). This adaptability highlights the flexibility of UNIQ.

When safety is critical, setting $\beta = 1$ ensures the most conservative policy, aligning with the objective of avoiding unsafe demonstrations. On the other hand, by varying $\beta$, one can tune the trade-off to achieve policies that are less safe but yield higher rewards, making UNIQ suitable for a range of scenarios depending on the desired safety-performance balance. This versatility demonstrates its practicality across different applications with varying safety requirements.

| | | DWBC | SafeDICE | UNIQ (0.01) | UNIQ (0.1) | UNIQ (0.3) | UNIQ (1.0) | UNIQ (3.0) |
|---|---|---|---|---|---|---|---|---|
| Point-Goal | Return | 26.9±0.1 | 27.0±0.1 | 25.1 ± 0.1 | 25.0 ± 0.1 | 24.5 ± 0.2 | 23.4±0.4 | 24.3 ± 0.2 |
| | Cost | 45.8±3.4 | 46.8±3.1 | 44.0 ± 3.6 | 40.5 ± 3.2 | 32.5 ± 3.2 | **27.1±3.0** | 30.8 ± 2.9 |
| Car-Goal | Return | 32.8±0.7 | 33.5±0.7 | 30.6 ± 0.5 | 30.1 ± 0.5 | 29.0 ± 0.6 | 27.9±0.8 | 28.9 ± 0.6 |
| | Cost | 47.4±3.8 | 50.5±4.0 | 46.4 ± 4.7 | 45.9 ± 4.2 | 41.2 ± 3.4 | **31.0±2.8** | 40.4 ± 3.4 |

Table 9: Comparison with different $\alpha$

## E.11 Full Numerical Experiment for Mujoco Velocity Tasks

We evaluate our method on two MuJoCo velocity tasks: Cheetah and Ant. we test the method with varying sizes of the undesired dataset, annotated as "env-UN= $\{1, 5, 10\}$" while the unlabeled dataset $\mathcal{D}^{\text{Mix}}$ is combined from 1600 high-cost and 400 low-cost trajectories. The detailed results are summarized in Table 10 and learning curves are shown in Figure 17 and Figure 18. Overall, increasing the size of the undesired dataset helps SafeDICE and DWBC achieve higher performance, while UNIQ reaches its peak performance with just a single undesired trajectory.

|  |  | DWBC | SafeDICE | UNIQ |
|---|---|---|---|---|
| Cheetah-UN=10 | Return | 3135.6±127.4 | 2841.9±56.1 | **2662.0±33.1** |
|  | Cost | 311.0±116.0 | 550.2±13.5 | **0.0±0.0** |
|  | CVaR | 897.7±10.0 | 682.2±14.4 | **0.0±0.0** |
| Cheetah-UN=5 | Return | 3430.9±107.5 | 2860.8±57.8 | **2661.2±29.7** |
|  | Cost | 578.8±89.6 | 553.9±25.3 | **0.0±0.0** |
|  | CVaR | 909.2±6.3 | 686.7±17.3 | **0.0±0.0** |
| Cheetah-UN=1 | Return | 3720.7±39.2 | 2910.0±61.8 | **2755.3±23.8** |
|  | Cost | 823.0±17.5 | 575.5±23.0 | **0.0±0.0** |
|  | CVaR | 916.4±5.0 | 702.2±20.0 | **0.0±0.0** |
| Ant-UN=10 | Return | 2225.0±759.3 | 2713.0±56.2 | **2850.5±177.5** |
|  | Cost | 470.5±162.8 | 439.7±57.7 | **15.2±10.8** |
|  | CVaR | 795.0±103.7 | 668.7±14.6 | **24.6±13.7** |
| Ant-UN=5 | Return | 2210.0±655.7 | 2727.4±49.8 | **2838.2±177.9** |
|  | Cost | 494.5±146.8 | 464.4±35.3 | **13.1±7.5** |
|  | CVaR | 805.7±16.4 | 671.2±16.0 | **22.1±10.5** |
| Ant-UN=1 | Return | 2259.4±653.8 | 2724.4±90.7 | **2841.4±214.9** |
|  | Cost | 507.5±147.8 | 506.5±40.3 | **16.9±7.1** |
|  | CVaR | 789.3±91.3 | 685.8±16.4 | **27.2±9.6** |

Table 10: Full comparison between UNIQ and other baselines in Mujoco-velocity domain. With decreasing of undesirable dataset size, the performance of DWBC and SafeDICE become worse. In contrast, UNIQ able to achieve highest performance with just a single undesirable trajectory.

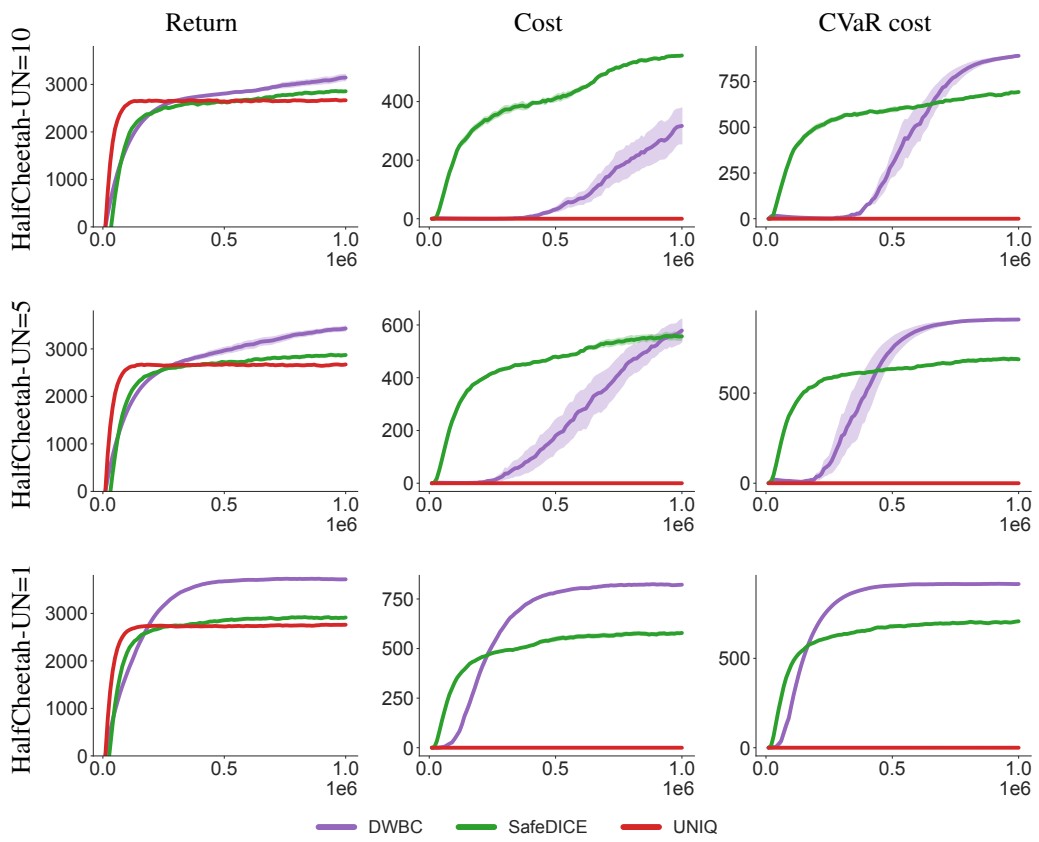

Figure 17: Cheetah task with unlabeled dataset(400-1600) and different undesired dataset.

## E.12 Performance with the Dataset Employed in the SafeDICE Paper

As we are using a different dataset from the SafeDICE dataset, we also provide a comparison with the dataset from SafeDICE paper. The detailed performance of the expert dataset is shown in Table 11:

|  | Point-Goal | Point-Button |
|---|---|---|
| Mean non-preferred demonstrations cost | 20.018 | 21.933 |
| Mean preferred demonstrations return | 19.911 | 8.286 |
| Mean non-preferred demonstrations cost | 107.977 | 166.099 |
| Mean preferred demonstrations return | 13.798 | 12.085 |

Table 11: SafeDICE dataset performance.

We mix 300 preferred demonstrations and 1200 non-preferred demonstrations for the unlabeled dataset and use 100 non-preferred demonstrations for the undesired dataset. The performance is shown in Figure 19. It is clearly that our method can achieve lower cost than SafeDICE.

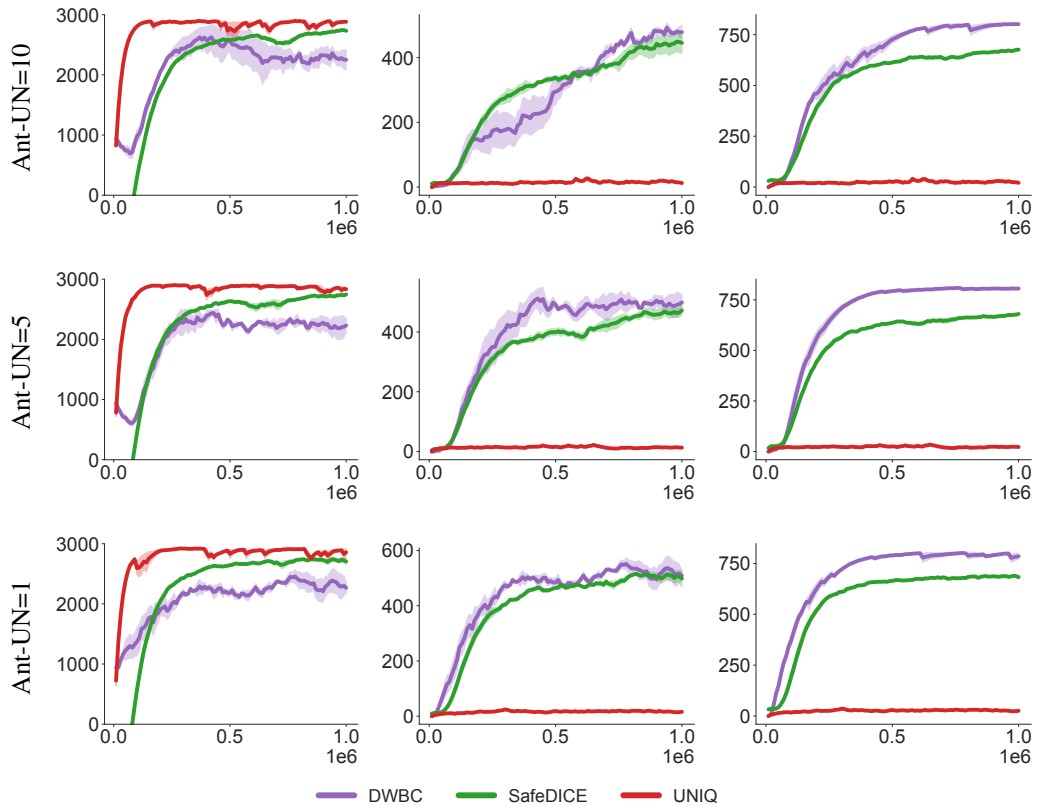

Figure 18: Ant task with unlabeled dataset(400-1600) and different undesired dataset.

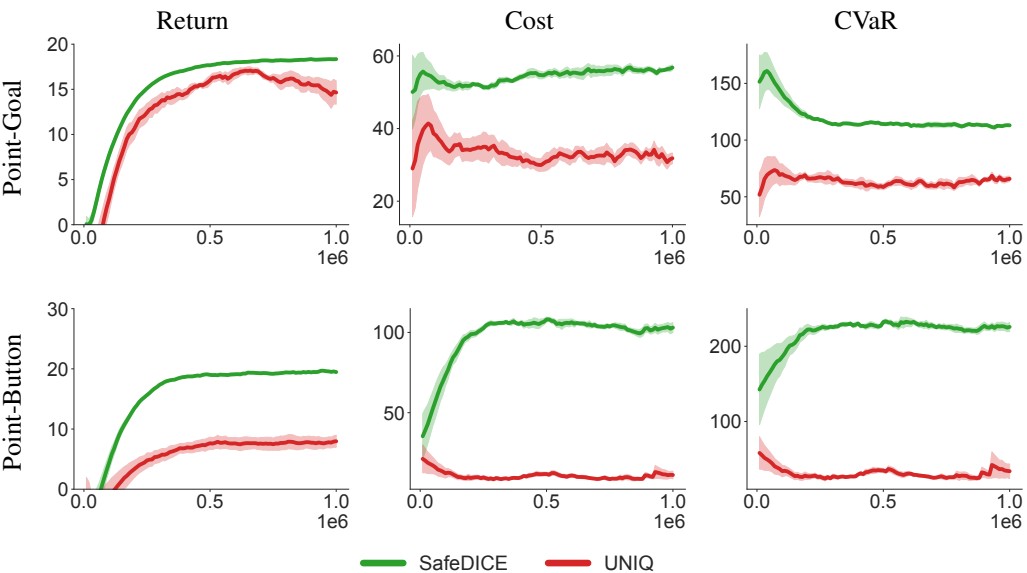

Figure 19: Comparison between UNIQ compared to SafeDICE in their dataset.

## E.13 Occupancy Ratio Estimation Analysis

To further address the reviewer's insightful comments regarding the accuracy of discriminator-based occupancy ratio estimation, we conduct an additional analysis to empirically examine how well our discriminator approximates the true occupancy ratio under a controlled setting.

**Motivation.** As noted by the reviewer, density ratio estimation using a discriminator can be sensitive when the underlying distributions of undesirable and unlabeled data differ significantly. However, it is also important to understand the behavior of the estimator when both datasets originate from the same distribution, where the ground-truth occupancy ratio $\tau^*(s, a) = 1$ for all $(s, a)$. This experiment allows us to isolate the intrinsic estimation accuracy of the discriminator, independent of distributional shift effects.

**Experimental setup.** We construct synthetic datasets by randomly sampling both undesirable and unlabeled trajectories from a shared data pool, ensuring identical underlying state-action distributions. We then train our discriminator following the same procedure described in our method, and compute the estimated occupancy ratio $\tau(s, a)$ for varying dataset sizes $|\mathcal{D}| \in \{10, 20, 50, 100, 300, 500, 1000\}$ trajectories. For each configuration, we report the mean and standard deviation of the estimated ratio across multiple random seeds.

**Results.** The results in Table 12 demonstrate that the estimated occupancy ratio $\tau$ converges rapidly toward the true value of 1 as the number of trajectories increases. When the dataset size exceeds roughly 300 trajectories, both the mean and standard deviation stabilize around $1.0 \pm 0.0$, indicating that the discriminator can accurately approximate the true occupancy ratio given a moderate amount of data. This suggests that the density ratio estimation subproblem in our framework is not a limiting factor for policy learning in practice.

| Sample size | 10 | 20 | 50 | 100 | 300 | 500 | 1000 |
|---|---|---|---|---|---|---|---|
| **Estimated** $\tau$ | $0.26 \pm 0.46$ | $0.34 \pm 1.33$ | $0.77 \pm 0.30$ | $0.85 \pm 0.57$ | $\mathbf{1.00 \pm 0.00}$ | $\mathbf{1.00 \pm 0.00}$ | $\mathbf{1.00 \pm 0.00}$ |

Table 12: Empirical evaluation of discriminator-based occupancy ratio estimation when the undesirable and unlabeled datasets are drawn from the same distribution. The true ratio is expected to be 1 for all $(s, a)$ pairs. The estimated ratio $\tau$ converges toward 1 as the number of trajectories increases.

## E.14 Ablation Study: Extreme Q-Learning

To evaluate the empirical impact of the Extreme Q-learning (XQL) update that contribute to the success of our method, we conducted an ablation study comparing XQL with alternative Q-function update strategies: Implicit Q-Learning (IQL) and Sparse Q-Learning (SQL).

Table 13 reports the results across four continuous control tasks. We observe that XQL consistently outperforms both IQL and SQL, indicating that the extreme Q-update is a key contributor to the performance gains reported in Section 5.2.

Table 13: Ablation study comparing different Q-function update strategies. Results are reported as mean $\pm$ standard deviation over 5 seeds.

| | Cheetah | Ant | Hopper | Walker2d |
|---|---|---|---|---|
| IQL | $3.8 \pm 2.7$ | $21.0 \pm 27.0$ | $57.2 \pm 12.8$ | $85.1 \pm 19.2$ |
| SQL | $54.2 \pm 5.1$ | $38.3 \pm 23.3$ | $53.9 \pm 17.3$ | $85.9 \pm 21.1$ |
| XQL | $\mathbf{75.7 \pm 6.8}$ | $\mathbf{104.4 \pm 10.5}$ | $\mathbf{73.5 \pm 20.6}$ | $\mathbf{105.9 \pm 4.0}$ |

These results demonstrate that the extreme Q-update substantially improves performance across all evaluated tasks, confirming its effectiveness as a critical component of our approach.

## E.15 significantly larger number of undesirable demonstrations

we conducted additional experiments using a significantly larger number of undesirable demonstrations to enable further analysis and comparison. The table 14 report the performance of UNIQ

and baseline methods on the Car-Goal and Point-Goal tasks, with up to 1000 undesirable demonstrations. Overall, the results show that UNIQ consistently achieves lower costs—indicating safer policies—while maintaining comparable returns to the baselines. This supports the robustness and effectiveness of UNIQ even in heavily imbalanced settings dominated by suboptimal data.

| Task | Method | $|\mathcal{D}^{\text{UN}}| =$ | 100 | 300 | 500 | 700 | 1000 |
|------|--------|------|-----|-----|-----|-----|------|
| **Point-Goal** | DWBC | Return | 26.9±0.1 | 26.6±0.2 | 26.4±0.3 | 26.5±0.4 | 25.6±0.3 |
| | | Cost | 45.8±3.4 | 39.1±2.6 | 35.9±3.1 | 36.1±2.7 | 33.2±2.1 |
| | SafeDICE | Return | 27.0±0.1 | 26.9±0.1 | 26.9±0.1 | 25.6±0.3 | 25.8±0.1 |
| | | Cost | 46.8±3.1 | 44.8±3.6 | 42.3±3.7 | 38.0±1.8 | 38.4±0.8 |
| | UNIQ | Return | 23.4±0.4 | 21.1±0.6 | 21.0±0.7 | 20.7±0.4 | 20.9±0.3 |
| | | Cost | 27.1±3.0 | 22.4±3.6 | 20.2±3.7 | 19.4±4.2 | 18.7±3.2 |
| **Car-Goal** | DWBC | Return | 32.8±0.7 | 28.8±1.5 | 25.6±2.2 | 24.1±1.8 | 23.9±2.3 |
| | | Cost | 47.4±3.8 | 37.8±3.1 | 30.6±3.4 | 23.5±3.6 | 19.9±4.9 |
| | SafeDICE | Return | 33.5±0.7 | 32.7±0.8 | 31.9±0.8 | 31.5±0.7 | 31.1±0.0 |
| | | Cost | 50.5±4.0 | 47.2±4.2 | 43.2±4.0 | 44.0±5.5 | 42.7±1.7 |
| | UNIQ | Return | 27.9±0.0 | 24.6±0.9 | 23.3±0.7 | 23.1±0.8 | 22.1±1.1 |
| | | Cost | 31.0±2.8 | 22.8±2.9 | 19.6±2.2 | 20.0±3.7 | 18.9±2.4 |

Table 14: Results for Point-Goal and Car-Goal tasks. Each cell reports mean ± std.

## E.16 Impact of Unlabeled Data Quality

To further analyze the robustness of our method, we conducted an additional experiment to investigate how the quality of the unlabeled dataset impacts performance. A key question is how our approach fares when the proportion of undesirable (random and medium) trajectories in the unlabeled dataset is substantially increased.

For this study, we evaluated our method on two MuJoCo tasks: Cheetah and Ant. We maintained the same undesirable dataset, which consists of 5 random and 5 medium trajectories. The unlabeled dataset was constructed by combining a varying number of random and medium trajectories (X) with the same 100 expert trajectories used in our main experiments. The total size of the unlabeled dataset is denoted as X:100, where X represents the combined count of random and medium trajectories, split equally between the two.

The results, as presented in Table 15, demonstrate the performance of SafeDICE, DWBC, and UNIQ as we increase the number of non-expert trajectories in the unlabeled dataset from 1000 to 3000.

| **Cheetah** | 1000:100 | 1500:100 | 2000:100 | 2500:100 | 3000:100 |
|---------|----------|----------|----------|----------|----------|
| SafeDICE | 3.1±0.9 | 5.1±5.2 | 5.2±4.0 | 4.3±6.1 | 2.4±3.2 |
| DWBC | 2.2±0.0 | 2.2±0.0 | 2.2±0.0 | 2.2±0.0 | 2.2±0.0 |
| UNIQ | 75.7±6.8 | 76.0±1.1 | 71.2±4.9 | 67.5±3.1 | 64.8±5.3 |

| **Ant** | 1000:100 | 1500:100 | 2000:100 | 2500:100 | 3000:100 |
|-----|----------|----------|----------|----------|----------|
| SafeDICE | 4.4±2.6 | 5.8±7.8 | 5.6±3.7 | 4.7±4.1 | 5.3±5.8 |
| DWBC | 84.6±12.4 | 84.9±1.9 | 84.4±3.8 | 83.4±4.3 | 84.3±6.8 |
| UNIQ | 104.4±10.5 | 105.2±9.0 | 100.0±7.2 | 100.7±7.3 | 99.5±8.8 |

Table 15: Performance evaluation on Cheetah and Ant tasks with varying sizes of unlabeled random and medium trajectories.

## E.17 Ablation Study: Sensitivity to Hyperparameters

While our method aims for minimal hyperparameter tuning, it is important to understand its sensitivity to key parameters. To address this, we conducted an ablation study focusing on the entropy regularization coefficient, $\beta$, which is a standard hyperparameter in maximum entropy frameworks.

The results of this study are presented in Table 16. Our findings are consistent with prior work on maximum entropy methods, showing that the performance of UNIQ remains stable across a reasonable range of $\beta$ values. For other standard hyperparameters, such as the regularization strength and the discriminator architecture, we adopted settings from prior work. These choices have been shown to be effective and well-validated, representing strong defaults for our setting.

| $\beta$ | 0.5 | 1 | 3 | 5 | 10 |
|---|---|---|---|---|---|
| Cheetah | 13.7±4.8 | 16.2±3.0 | 74.4±3.9 | **75.3±4.3** | 6.3±5.7 |
| Ant | 90.8±20.0 | **104.3±1.6** | 57.2±19.3 | 26.5±4.2 | 24.6±8.2 |
| Hopper | 7.3±3.9 | 4.0±1.8 | 43.4±20.9 | 73.4±16.8 | **75.1±11.0** |
| Walker | 20.2±14.7 | 34.5±20.0 | 80.6±15.2 | **105.9±4.0** | 56.8±4.6 |

Table 16: Performance of UNIQ across different values of the entropy regularization coefficient $\beta$. The results show that performance is stable within a reasonable range of values. The best performance for each task is highlighted in bold.

As shown in the table, the optimal value for $\beta$ varies depending on the task. For instance, the Ant environment performs best with a smaller $\beta$ of 1, while the Hopper environment benefits from a larger $\beta$ of 10. Despite this variation, the performance remains high for a range of values around the optimum for each task (e.g., $\beta \in [3, 5]$ for Cheetah and Walker). This indicates that while $\beta$ is an important parameter, our method is not overly sensitive to its precise value and does not require exhaustive tuning to achieve strong results.

