# OpenReview forum: "No Experts, No Problem: Avoidance Learning from Bad Demonstrations"
_NeurIPS.cc/2025/Conference — NeurIPS 2025 poster_

### Official Review · Reviewer_v5LQ · 2025-06-22

**Clarity:** 3
**Significance:** 2
**Originality:** 2
**Rating:** 4
**Confidence:** 3

**Summary:**

The paper introduces UNIQ (Undesired Demonstrations driven Inverse Q-learning), a method for offline imitation learning in settings where two types of data are available: a small set of undesirable demonstrations, and a large pool of unlabeled demonstrations that may include both good and bad behaviors. The objective is to learn a policy that avoids learning the undesirable behaviors while still acquiring task-relevant skills from the unlabeled data.

UNIQ builds on the IQ-Learn framework by inverting its objective: rather than maximizing reward on expert trajectories as in standard imitation learning, UNIQ instead minimizes reward on the undesirable trajectories. To leverage the much larger unlabeled dataset, UNIQ introduces an occupancy correction mechanism where it reweighs the learning objective using the likelihood ratio of how likely a state-action pair in the unlabeled set originates from the undesirable distribution.

Experiments are conducted on 8 environments from D4RL and Safety-Gym. Results show the UNIQ achieves lower safety costs or constraint violations and high returns compared to several baselines.

**Questions:**

- Have you tried more diverse data compositions for experiments? For example, some medium behaviors in undesirable set, and random+expert in unlabeled set.
- What is the empirical impact of the Extreme Q-learning in Section 4.3? Do you have ablations?

**Ethical Concerns:**

["NO or VERY MINOR ethics concerns only"]

**Limitations:**

The authors do provide a limitations section. I would also strongly encourage them to think how difficult it is to capture undesirable demonstrations. People always talk of the difficulty of curating an expert set, but I think it is also very challenging to capture really "bad" demonstrations. Some clarity on practical curation would be welcome.

I don't believe the method has any distinctive negative societal impact beyond those that exist for ML methods.

**Quality:**

2

**Strengths And Weaknesses:**

**Strengths**
- The paper addresses a practically important problem, as demonstrated by the examples provided for autonomous driving.
- Flipping the IQ-Learn objective leads to a loss of convexity with respect to the Q-function, which could make optimization unstable. The authors address this with a separately learned value function, which I found to be interesting.
- The paper is well-written and clearly organized.

**Weaknesses**

1. Experiments:
- Dataset creation: For the D4RL environments, the undesirable dataset is constructed by combining trajectories from the random and medium levels. The unlabeled dataset is formed by combining trajectories from all three levels: random, medium, and expert. For the Safety-Gym environments, two policies are first trained: a constrained policy that respects safety constraints, and an unconstrained policy that optimizes for return while ignoring safety. Data collected from the unconstrained policy is used to create the undesirable dataset, while data from both policies is combined to form the unlabeled dataset. This results in the unlabeled data having a clear bimodal distribution. Furthermore, for the D4RL setup, the space of undesired behavior is extremely large and essentially encompasses everything except for expert behavior.
- Baselines: Building on the above point, DWBC should solve this problem quite easily. DWBC originally learns a discriminator to distinguish between expert and non-expert data using PU learning. Thus, one can easily adapt it by treating unsafe data as the "positive" data in PU learning and then performing BC using inverse weighting instead of direct weighting. However, the authors choose the unlabeled dataset as the "positive" class which is confusing and a much weaker baseline.
- Ablation: The impact of adding the extra value function is not analyzed.

2. The likelihood estimation is done by training a binary classifier. This will lead to issues since the the negative class (unlabeled set) contains the positive class (undesirable set).

3. The paper is slightly incremental, since it is basically flips the objective of IQ-Learn.

---

> ### Author Rebuttal · Authors · 2025-07-30
>
> *We thank the reviewer for the insightful comments. Below, we provide our detailed responses to each of your questions and concerns.*
>
> ---
>
> > Baselines: Building on the above point, DWBC should solve this problem quite easily. DWBC originally learns a discriminator...
>
> We thank the reviewer for the insightful comment. While we had previously adapted DWBC to better fit our setting, the approach you suggested provides a valuable alternative perspective. To address this, we conducted an additional experiment using the version you suggested, and we report the results below. Overall, the results indicate that our adapted version of DWBC generally achieves better performance.
> | | Cheetah | Ant | Hopper | Walker2d |
> |---|---|---|---|---|
> | Suggested DWBC (return) | 4.7±1.4 | 38.7±2.5 | 12.2±3.9 | 5.9±7.8 |
> | Our adapted DWBC (return)| 2.2±0.0 | 84.6±12.4 | 14.8±15.7 | 38.8±24.7 |
>
> | | Point-Goal | Car-Goal | Point-Button | Car-Button |
> |---|---|---|---|---|
> | Suggested DWBC | Return = 27.1±0.1; Cost = 47.9±0.3 | Return = 35.4±0.5; Cost = 54.8±2.3 | Return = 24.5±0.8; Cost = 142.4±5.0 | Return = 18.2±0.5; Cost = 271.4±31.2 |
> | Our adapted DWBC | Return = 26.9±0.1; Cost = 45.8±3.4 | Return = 32.8±0.7; Cost = 47.4±3.8 | Return = 17.2±0.9; Cost = 123.5±14.4 | Return = 17.1±1.0; Cost = 249.2±20.9 |
>
> > The likelihood estimation is done by training a binary classifier. This will lead to issues since the the negative class (unlabeled set) contains the positive class (undesirable set).
>
> We thank the reviewer for the insightful comments. In theory, the discriminator is trained to estimate an occupancy ratio that allows us to leverage unlabeled samples while remaining consistent with the original learning objective. In practice, the discriminator tends to assign higher values to τ(s,a) when the state-action pair appears more frequently in the undesirable dataset than in the unlabeled one, and lower values otherwise. This estimated ratio is directly used in our learning objective (Eq. 6) to guide the policy away from undesirable behaviors and toward better policies. Therefore, the presence of undesirable samples within the unlabeled dataset does not pose a problem for our method.
>
> > The paper is slightly incremental, since it is basically flips the objective of IQ-Learn.
>
> We thank the reviewer for the comment. While our approach builds on the structure of inverse Q-learning and, more broadly, the maximum entropy (MaxEnt) framework, it introduces a novel perspective by explicitly focusing on learning from undesirable behaviors—a setting that differs significantly from traditional imitation learning methods, which rely on expert or near-expert demonstrations. Although our objective mirrors the MaxEnt framework in form, it is reconstructed in the opposite direction, and many theoretical properties from MaxEnt no longer apply. This shift requires new analysis and empirical validation. For these reasons, we believe our work offers a meaningful contribution that advances the field in a new and underexplored direction.
>
> > Have you tried more diverse data compositions for experiments? For example, some medium behaviors in undesirable set, and random+expert in unlabeled set.
>
> We thank the reviewer for the question. To clarify, our datasets are already diverse: the undesirable data includes both medium and random demonstrations, while the unlabeled dataset contains a mix of random, medium and expert demonstrations. In the experiments presented in both the main paper and the appendix, we include several ablation studies—specifically, Section E.1 shows results when more random and medium demonstrations are added to the undesirable set, and Sections E.2 and E.3 explore the effect of adding more random and expert demonstrations to the unlabeled dataset. We hope these experiments address your concern.
>
> > What is the empirical impact of the Extreme Q-learning in Section 4.3? Do you have ablations?
>
> We thank the reviewer for the constructive comment. To evaluate the impact of our extreme Q-update, we conducted additional experiments comparing different Q-function update strategies, including Implicit Q-Learning (IQL) and Sparse Q-Learning (SQL). The results are reported below, showing that Extreme Q-learning (XQL)  achieves the best performance as.
>
> | | Cheetah | Ant | Hopper | Walker2d |
> |---|---|---|---|---|
> | IQL | 3.8±2.7 | 21.0±27.0 | 57.2±12.8 | 85.1±19.2 |
> | SQL | 54.2±5.1 | 38.3±23.3 | 53.9±17.3 | 85.9±21.1 |
> | XQL | 75.7±6.8 | 104.4±10.5 | 73.5±20.6 | 105.9±4.0 |
>
> ---
>
> *We hope the above responses satisfactorily address your concerns. We will update the paper to include the additional experiments  and discussion described above. If you have any further comments or questions, we would be happy to address them.*

---

> ### Author Response · Authors · 2025-08-06
> **Reply to Reviewer v5LQ**
>
> *We thank the reviewer for the response and raising important points for further discussion.*
>
> ---
>
> > For DWBC, do you have any insights as to why this performs worse - is it because of the proportion parameter $\eta$ in the PU classifier not being representative of the data mixture?
>
> We appreciate the reviewer's active engagement and insightful follow-up question. While we don't have a definitive explanation for why the **suggested DWBC** variant underperforms our **adapted version**, we can offer the following interpretation, focusing on the core mechanisms and their suitability for different learning scenarios.
>
> ### **The Problem with Aggressive Weight Separation**
>
> Both DWBC variants attempt to assign diverse weights to samples from **bad** and **unlabeled** datasets to train a policy using a behavior cloning (BC) objective. The goal is for the policy to follow unlabeled data while avoiding bad demonstrations.
>
> However, when the **unlabeled dataset is of low quality**—which is often the case in our setting—aggressively assigning a "high" weight to unlabeled samples and a "low" weight to bad samples can be problematic.
>
> - The **suggested DWBC** variant tries to assign diverse weights to distinguish between the two datasets as much as possible. This aggressive separation works well in a **learning-from-expert setting**, where the difference between expert and unlabeled data is distinct, making it effective to heavily favor expert samples.
> - In contrast, the way **PU learning** is used in our **adapted DWBC** helps mitigate the divergence between the weights assigned to the two datasets, leading to more stable and reliable outcomes, especially when the quality of bad and unlabeled data is similar.
>
> ### **Our UNIQ Framework vs. DWBC**
>
> This reasoning also highlights the key difference between our UNIQ framework and the DWBC variants.
>
> - **DWBC** relies on a weighting scheme to differentiate between bad and unlabeled data, a mechanism that can fail when the datasets are of similar quality.
> - **UNIQ** avoids this issue entirely by learning to directly **avoid bad samples** while only using unlabeled data to improve sample efficiency. This approach is more robust because it doesn't depend on a potentially unreliable weighting scheme.
>
> Regarding the parameter $\eta$, we used a default value of $\eta=0.5$ across all experiments. We also tried with higher values of $\eta$. The performance gap stems from the core mechanism of the original DWBC, which is not well-suited for the learning-from-bad-demonstrations setting, rather than from a mismatch in the proportion parameter.
>
> > For the data compositions, I was implying towards more discrepancy between unlabeled and undesired set, rather than scaling: some medium behaviors in undesirable set, and random+expert in unlabeled set. The aim is to investigate whether the method works even when all "types" of undesired data is not enumerated.
>
> We thank the reviewer for the additional feedback, which gave us another valuable opportunity to address your concern. In response to your suggestion, we conducted an experiment using your proposed setting: medium behaviors placed in the undesirable dataset and a mix of random+expert trajectories in the unlabeled set.
>
> The results, shown below, indicate that UNIQ—as well as other DWBC-based methods—struggle to learn effective policies in this configuration. We believe the main reason is that avoiding medium demonstrations in the undesirable dataset may unintentionally steer the policy away not only from suboptimal behavior but also towards random behavior present in the unlabeled set. This misdirection undermines the learning process and leads to degraded performance.
>
> | | Cheetah | Ant | Hopper | Walker2d |
> |---|---|---|---|---|
> | DWBC | 2.2±0.0 | 27.4±3.7 | 1.3±0.1 | 0.2±0.2 |
> | SafeDICE | -0.0±0.0 | -3.4±0.3 | 3.2±4.3 | 0.0±0.0 |
> | UNIQ | 2.0±0.2 | 28.2±0.6 | 1.8±2.3 | 0.1±0.0 |
>
> ---
>
> *We hope this discussion and the accompanying experiment address your remaining concerns. We will incorporate these results and clarifications into the final version of the paper. Please feel free to let us know if you have any further questions or suggestions.*

---

### Official Review · Reviewer_ffqG · 2025-06-23

**Clarity:** 3
**Significance:** 2
**Originality:** 3
**Rating:** 5
**Confidence:** 5

**Summary:**

The paper introduces a new imitation learning setting with the goal of both avoiding bad demonstrations and learning from unlabeled demonstrations. They present a non-adversarial method (UNIQ) to do this by combining 3 ideas from past work: IQ-Learn, Extreme Q-Learning, and occupancy ratio estimation. They demonstrate that UNIQ outperforms baselines in constrained and unconstrained continuous control settings.

**Questions:**

Minor
1. I'm not sure what point the toy MDP setting is supposed to make: wouldn't minimizing KL against a mixing baseline like SafeDICE (which down-weights p1) still upweight p2 and p3?
2. Most of the baselines seem incapable of learning in the unconstrained setting. Do you have an explanation for why this happens and what their limiting factor is?
3. Just to clarify: for Proposition 4.1 and Section 4.3, is the optimal V being relative to the behavior policy a property of the offline setting?
4. Do you have a sense of whether the cooperative nature of the setting allow you to do better than the expert?
5. Figure 3 seems to demonstrate that UNIQ scales better with expert demos vs other methods. Do you have an explanation for this behavior?
6. Are there results for the data scaling experiments (expert and undesirable demos) with larger amounts of data? Right now, the ablation only spans 1 OOM.
7. This is out of scope of the paper, but do you have any thoughts on how UNIQ scales in the online setting?

Major
1. Re. weakness 1: I think the setting is novel, but the scope is pretty narrow to the offline, data-constrained setting, and I have some concerns of broader significance. I'm willing to change my mind on this, given discussion and supporting experiments. For example, one such argument could be: there are settings where  it's easier to classify bad behavior than good behavior and UNIQ gives significant gains. Showing this would require 1. Evidence of such a setting where this gap exists 2. Empirical evidence that UNIQ scales better with undesirable demos than baselines in this setting. For example, Figure 4 (Point-Goal) doesn't seem to really show this as the baselines seem to maintain better return with decreasing cost (and they might cross if the x-axis was extended). I'm open to other arguments and happy to discuss.
2. Re. weakness 2: I have some concerns about estimating the occupancy measure using a discriminator. Although the idea is pretty classic, density ratio estimation in this way can struggle to accurately model the scale of the ratio when the distributions are far apart. I wonder if this is an issue if the undesirable and unlabeled distributions grow further apart. I'd be very interested in seeing an ablation doing some preliminary science for this. I believe this would be a valuable contribution for understanding the empirical properties of algorithmic primitives for estimating occupancy ratios. I think you could do this in the safety constrained setting where the oracle occupancy ratio is known since you have likelihoods of the PPO-Lagrangian and PPO policies. The main experiment would then show that NCE is sufficient to learn a "good estimator" (as measured by some divergence against the oracle) and goodness of estimation scales favorably with the amount of undesirable data. Note: I view this as a valuable contribution for understanding, even if the experiment shows that the discriminator doesn't scale well with more undesirable data.
3. I'm willing to increase my rating to 5 if both points are addressed adequately.

**Ethical Concerns:**

["NO or VERY MINOR ethics concerns only"]

**Final Justification:**

I updated my score from a 4 to a 5 after rebuttal. I had 2 main concerns: 1. scope of the work and scaling behavior of UNIQ with respect to undesirable demos 2. estimation of occupancy ratios. The authors did a reasonable job of addressing both concerns. For 1, they argue that the proposed setting is relevant to several real-world safety-critical applications and show additional results for positive behavior when scaling the number of undesirable demos. For 2, they provided 2 different sanity check experiments that the estimated occupancy ratio is reasonable. Both of my main concerns were adequately addressed, so I raised my score.

**Limitations:**

Yes

**Quality:**

3

**Strengths And Weaknesses:**

Strengths
1. The overall presentation of the work is clear. The authors do a good job of summarizing major related work, motivating their setting, and iteratively building up the core parts of their method.
2. The core set of results (Table 1) are convincingly better across both unconstrained and constrained RL settings. UNIQ is the only method that matches (or comes close) to expert performance across all settings. The authors do a reasonable job of doing the science of ablating data scaling, see Questions for some more comments on this point.
3. The method mostly combines well-known ideas (IQ-Learn, Extreme Q-learning, occupancy ratio estimation), but it's well-motivated and applies these ideas to a fresh setting. This combination distinguishes the method and setting from past work.

Weaknesses
1. My main concern is that the scope is pretty narrow to the offline, data-constrained setting, and I have some concerns of broader significance.
2. I have a secondary concern about the method for estimating the occupancy measure ratio using a discriminator and would like to see rigorous experiments characterizing this.
3. I'm willing to update my score based on rebuttal, see Questions for specifics.

---

> ### Author Rebuttal · Authors · 2025-07-30
>
> *We thank the reviewer for the insightful comments. Below, we provide our detailed responses to each of your questions and concerns.*
>
> ---
>
> > I'm not sure what point the toy MDP setting is supposed to make: wouldn't minimizing KL against a mixing baseline like SafeDICE (which down-weights p1) still upweight p2 and p3?
>
> The purpose of the toy MDP is not to compare against other baselines, but to illustrate that our learning objective can return near-optimal policies when learning from undesirable demonstrations. We believe that methods like SafeDICE could also do the same in this toy example.
>
> > Most of the baselines seem incapable of learning in the unconstrained setting. Do you have an explanation for why this happens and what their limiting factor is?
>
> Among all the baselines, only SafeDICE is explicitly designed to learn from undesirable demonstrations, and it primarily focuses on constrained settings. As discussed in our paper, SafeDICE ‘s performance is highly sensitive to the quality of the unlabeled dataset. In contrast, our approach is theoretically designed to be less dependent on the quality of the unlabeled data, offering greater robustness in challenging settings.
>
> > Just to clarify: for Proposition 4.1 and Section 4.3, is the optimal V being relative to the behavior policy a property of the offline setting?
>
> Yes, this is a common property of the offline setting, where data is assumed to be sampled from a behavior policy. This assumption is widely adopted in both offline RL and IL.
>
> > Do you have a sense of whether the cooperative nature of the setting allow you to do better than the expert?
>
> In our setting, outperforming the expert cannot be guaranteed and is inherently challenging, as we do not have access to expert demonstrations for guidance. Instead, the cooperative nature of our approach contributes to a more stable training process and helps steer learning toward a good policy.
>
> > Figure 3 seems to demonstrate that UNIQ scales better with expert demos vs other methods. Do you have an explanation for this behavior?
>
> We thank the reviewer for the comment. Compared to other baselines, our method can better leverage expert demonstrations within the unlabeled data. As shown in our learning objective (Eq. 6), when more expert demonstrations are present, the algorithm assigns lower τ(s,a) values to those expert-like pairs, encouraging higher rewards for them and ultimately leading to a better policy.
>
> > Are there results for the data scaling experiments (expert and undesirable demos) with larger amounts of data? Right now, the ablation only spans 1 OOM.
>
> We thank the reviewer for the comment. In our current ablation study (Section E.3), we vary the number of expert demonstrations from 50 to 300, which we believe reflects a realistic and practical range in scenarios where expert data is typically limited.
>
> > This is out of scope of the paper, but do you have any thoughts on how UNIQ scales in the online setting?
>
> We thank the reviewer for the question. While the current framework could be extended to the online setting, making it effective would require substantial effort, including adapting the learning objective and regularization, as well as benchmarking against a new set of baselines. We consider this a valuable and promising direction for future work.
>
> > Re. weakness 1: I think the setting is novel, but the scope is pretty narrow to the offline, data-constrained setting,...
>
> We thank the reviewer for the constructive feedback and the helpful suggestion. Our work indeed focuses on the offline, data-constrained setting, which we agree is narrower in scope compared to more general-purpose IL frameworks. However, we believe this setting is both practically relevant and novel, particularly in safety-critical applications (e.g., robotics, autonomous driving, or healthcare), where undesirable behaviors—such as collisions or failures—are easier to collect or identify than expert demonstrations. In such cases, it is often significantly more feasible to label or log what not to do, rather than gather consistent expert behavior.
>
> > Empirical evidence that UNIQ scales better with undesirable demos than baselines in this setting. For example, Figure 4 (Point-Goal) ...
>
> We thank the reviewer for raising this valuable point. To address it, we conducted additional experiments using a significantly larger number of undesirable demonstrations to enable further analysis and comparison. The tables below report the performance of UNIQ and baseline methods on the Car-Goal and Point-Goal tasks, with up to 1000 undesirable demonstrations. Due to NeurIPS rebuttal format restrictions, we are unable to include figures, so we report only the mean and std of the final scores. Overall, the results show that UNIQ consistently achieves lower costs—indicating safer policies—while maintaining comparable returns to the baselines. This supports the robustness and effectiveness of UNIQ even in heavily imbalanced settings dominated by suboptimal data.
>
>
> **Point-Goal**
>
> | $\|\mathcal{D}^{UN}\|$ | 100 | 300 | 500 | 700 | 1000 |
> |---|---|---|---|---|---|
> | DWBC | Return = 26.9±0.1; Cost = 45.8±3.4 | Return = 26.6±0.2; Cost = 39.1±2.6 | Return = 26.4±0.3; Cost = 35.9±3.1 | Return = 26.5±0.4; Cost = 36.1±2.7 | Return = 25.6±0.3; Cost = 33.2±2.1 |
> | SafeDICE | Return = 27.0±0.1; Cost = 46.8±3.1 | Return = 26.9±0.1; Cost = 44.8±3.6 | Return = 26.9±0.1; Cost = 42.3±3.7 | Return = 25.6±0.3; Cost = 38.0±1.8 | Return = 25.8±0.1; Cost = 38.4±0.8 |
> | UNIQ | Return = 23.4±0.4; Cost = 27.1±3.0 | Return = 21.1±0.6; Cost = 22.4±3.6 | Return = 21.0±0.7; Cost = 20.2±3.7 | Return = 20.7±0.4; Cost = 19.4±4.2 | Return = 20.9±0.3; Cost = 18.7±3.2 |
>
> **Car-Goal**
>
> | $\|\mathcal{D}^{UN}\|$  | 100 | 300 | 500 | 700 | 1000 |
> |---|---|---|---|---|---|
> | DWBC | Return = 32.8±0.7; Cost = 47.4±3.8 | Return = 28.8±1.5; Cost = 37.8±3.1 | Return = 25.6±2.2; Cost = 30.6±3.4 | Return = 24.1±1.8; Cost = 23.5±3.6 | Return = 23.9±2.3; Cost = 19.9±4.9 |
> | SafeDICE | Return = 33.5±0.7; Cost = 50.5±4.0 | Return = 32.7±0.8; Cost = 47.2±4.2 | Return = 31.9±0.8; Cost = 43.2±4.0 | Return = 31.5±0.7; Cost = 44.0±5.5 | Return = 31.1±0.0; Cost = 42.7±1.7 |
> | UNIQ | Return = 27.9±0.0; Cost = 31.0±2.8 | Return = 24.6±0.9; Cost = 22.8±2.9 | Return = 23.3±0.7; Cost = 19.6±2.2 | Return = 23.1±0.8; Cost = 20.0±3.7 | Return = 22.1±1.1; Cost = 18.9±2.4 |
>
> > Re. weakness 2: I have some concerns about estimating the occupancy measure using a discriminator. Although the idea is pretty classic, ... I believe this would be a valuable contribution for understanding the empirical properties of algorithmic primitives for estimating occupancy ratios.
>
> We thank the reviewer for the valuable and insightful comment. Regarding the accuracy of occupancy ratio estimation, we fully agree that it can be inaccurate when datasets are small in size—an issue common in practice. However, as shown in our experiments, our algorithm remains robust and effective even with limited undesirable data. The key reason is that precise occupancy ratio estimation is not strictly required for good policy learning. In our main training objective (Equation 6), the discriminator encourages higher τ(s,a) values for state-action pairs that appear more frequently in the undesirable dataset, and lower values otherwise. This naturally drives the learned Q-function and policy to assign lower rewards r^Q(s,a) to undesirable behaviors, guiding the agent toward better actions. This behavior aligns with findings from prior work such as GAIL and DICE-based methods, which also demonstrate robustness under noisy ratio estimates.
>
> > I think you could do this in the safety constrained setting where the oracle occupancy ratio is known since you have likelihoods of the PPO-Lagrangian and PPO policies.
>
> We thank the reviewer for the insightful  suggestion. However, we are currently unsure how to compute the oracle occupancy ratio using PPO or PPO-Lagrangian policies. Estimating this ratio would typically require access to both undesirable and unlabeled datasets, along with accurate sampling from the target policy's occupancy distribution. In our case, we are not sure how to use PPO policies to get  a reliable estimate of the ratio. If the reviewer could clarify the intended procedure, we would be happy to explore it further and include the results.
>
> > The main experiment would then show that NCE is sufficient to learn a "good estimator"...
>
> To further address your concern, we conducted an additional experiment to evaluate the estimation of the occupancy ratio τ(s,a) across varying dataset sizes. Due to NeurIPS rebuttal policy restrictions, we are unable to include figures, so we report only the mean and standard deviation of the estimated ratios across the (s,a) space. The results generally indicate that the estimation of the occupancy ratio becomes stable as the dataset size grows. In the revised version of the paper, we will include detailed distribution plots and provide further analysis to strengthen the discussion around this estimation.
>
> | $\|\mathcal{D}^{MIX}$\| | 50 | 100 | 200 | 500 | 1000 | 2000 | 5000 |
> |---|---|---|---|---|---|---|---|
> | $\tau$ | 0.60±0.20 | 0.61±0.19 | 0.60±0.19 | 0.61±0.20 | 0.61±0.20 | 0.62±0.20 | 0.60±0.21 |
>
> | $\|\mathcal{D}^{UN}\|$| 1 | 5 | 10 | 25 | 50 | 100 | 500 | 1000 |
> |---|---|---|---|---|---|---|---|---|
> | $\tau$ | 0.68±0.19 | 0.61±0.19 | 0.62±0.23 | 0.60±0.20 | 0.59±0.21 | 0.61±0.20 | 0.61±0.18 | 0.61±0.19 |
>
> ---
>
> *We hope the above responses satisfactorily address your concerns. We will update the paper to include the additional experiments  and discussion described above. If you have any further comments or questions, we would be happy to address them.*

---

> > ### Comment · Reviewer_ffqG · 2025-08-05
> >
> > Thank you for the detailed rebuttal and taking the time to run additional experiments! A few more comments:
> > 1. The updated undesirable demo scaling results look reasonable. I encourage the authors to include them in the final version of the paper.
> > 2. I agree that precise occupancy ratio estimation isn't a requirement for policy learning. It still seems valuable to have a sense of how hard the density ratio estimation problem is and how robust the overall learning procedure is to this subproblem. The oracle occupancy ratio setting I previously suggested was roughly: use the PPO-Lagrangian policy to rollout a large dataset of trajectories (and keep track of likelihoods) and treat these as expert / good, use the unconstrained PPO policy in the same way and treat them as bad trajectories, and measure how well you learn the ratio over these datapoints since you have access to generative data distributions. It's a bit late to run an actual experiment for this, but I'd be interested in the authors' thoughts.
> > 3. I appreciate the additional experiment on density estimation. The analysis would be more compelling if you had a way of comparing the accuracy of your estimated ratios to the true ratios (which the above setting would give one way of doing).

---

> > > ### Author Response · Authors · 2025-08-06
> > > **Reply to Reviewer ffqG**
> > >
> > > *We thank the reviewer for the response and raising important points for further discussion.*
> > >
> > > ---
> > >
> > > > 1. The updated undesirable demo scaling results look reasonable. I encourage the authors to include them in the final version of the paper.
> > >
> > > We thank the reviewer for the helpful suggestion. We will include these additional experiments, along with other valuable feedback from the reviewers, in the final version of the paper.
> > >
> > > > 2. The oracle occupancy ratio setting I previously suggested was roughly: use the PPO-Lagrangian policy to rollout a large dataset of trajectories (and keep track of likelihoods) and treat these as expert / good, use the unconstrained PPO policy in the same way and treat them as bad trajectories, and measure how well you learn the ratio over these datapoints since you have access to generative data distributions. It's a bit late to run an actual experiment for this, but I'd be interested in the authors' thoughts.
> > >
> > > We thank the reviewer for the insightful clarification, which we find valuable and thought-provoking. However, we believe it remains challenging to estimate a ground-truth occupancy ratio in this way for several reasons:
> > >
> > > - **Difficulty obtaining optimal policies:** In practice, acquiring truly optimal policies is difficult and typically requires a large amount of high-quality data. In our setting, it is not guaranteed that PPO or PPO-Lag will yield optimal policies that accurately reflect the true data distribution. Even if such policies were obtained, evaluating their optimality or measuring the gap to a true optimal policy remains non-trivial.
> > > - **Challenges in constrained RL:** Solving constrained RL problems is significantly harder than their unconstrained counterparts. Obtaining even near-optimal solutions with algorithms like PPO-Lag can be unreliable, making them an impractical reference for estimating the occupancy ratio.
> > > - **Estimating occupancy ratios:** Even assuming access to optimal policies, computing the corresponding occupancy ratios accurately is itself a difficult task. To our knowledge, there are no direct and unbiased methods to recover these ratios in a reliable way.
> > >
> > > In summary, while we appreciate and agree that this is an interesting direction worth exploring, we find it infeasible to implement within the scope of our current work and believe it would require further dedicated investigation.
> > >
> > > > 3. I agree that precise occupancy ratio estimation isn't a requirement for policy learning. It still seems valuable to have a sense of how hard the density ratio estimation problem is and how robust the overall learning procedure is to this subproblem.
> > >
> > > We thank the reviewer for the thoughtful suggestion. After further investigation, we identified a more feasible approach to evaluate the accuracy of our discriminator’s occupancy ratio estimation.
> > >
> > > Our method leverages the insight that if the bad and unlabeled datasets are drawn from the same distribution, the ground-truth occupancy ratio should be $\tau(s,a) = 1$ for all $(s,a)$. Based on this, we conducted an experiment where we randomly sampled bad and unlabeled trajectories from a shared dataset and estimated the occupancy ratio across different dataset sizes.
> > >
> > > The results (reported below) show that the estimated ratio approaches 1 as the number of trajectories increases, with both the mean and standard deviation stabilizing. This indicates that a sample size of around 300 trajectories is generally sufficient to approximate the ground-truth ratio reliably.
> > >
> > > | Sample size | 10 | 20 | 50 | 100 | 300 | 500 | 1000 |
> > > |---|---|---|---|---|---|---|---|
> > > | $\tau$ | 0.26±0.46 | 0.34±1.33 | 0.77±0.30 | 0.85±0.57 | 1.0±0.0 | 1.0±0.0 | 1.0±0.0 |
> > >
> > > ---
> > >
> > > *We hope this discussion and the accompanying experiment address your remaining concerns. We will incorporate these results and clarifications into the final version of the paper. Please feel free to let us know if you have any further questions or suggestions.*

---

> > > > ### Comment · Reviewer_ffqG · 2025-08-06
> > > >
> > > > Thank you for the additional experiment re. estimating occupancy ratios and the discussion. I'll update my score to a 5. Please include these experiments in an updated version of the paper.

---

> ### Author Response · Authors · 2025-08-07
>
> We thank the reviewer for maintaining a positive view of our work and for the valuable feedback, particularly regarding the estimation of the occupancy ratio, which has helped us significantly improve the paper. We will definitely incorporate all the discussions and additional experiments into the final version.

---

### Official Review · Reviewer_nyTV · 2025-07-01

**Clarity:** 3
**Significance:** 2
**Originality:** 2
**Rating:** 4
**Confidence:** 3

**Summary:**

This paper addresses the challenge of learning avoidant behaviors in offline imitation learning without expert demonstration. It proposes UNIQ, a novel framework that leverages undesirable demonstrations and unlabeled data to learn strategies to avoid undesirable behavior. Starting from the training goal based on maximum entropy, UNIQ reformulates the problem as a cooperative inverse Q-learning task and integrates unlabeled data for an effective strategy for offline training. Experiments on Mujoco and Safety-Gym have shown that UNIQ outperforms other baselines when given a set of undesirable demonstrations.

**Questions:**

- If the probability of the demonstration appearing in undesirable demonstrations is less than the probability of appearing in unlabeled data, then is the optimization direction of Equation 4 reversed, and have you done any relevant experiments? For example, you can simply duplicate two copies of the undesirable demonstrations or add more to the unlabeled data.
- At present, the number of experts in the setup is relatively small. Is there an experiment with a larger number of experts, and will it be better than other methods?
- Have you tried adding optimal demonstrations and undesirable demonstrations to train together, and will the results of UNIQ be better?
- The cost metric of the Car-Button in table 1 is much lower than the UNIQ on the LS-IQ method, but it shows that the UNIQ is the best, is it a recording error?
- What are the advantages of the method proposed in this paper compared to confidence based imitation learning methods, such as 2IWIL, CAIL, WGAIL, PN-GAIL [1-4].
- Has the time spent in each method been recorded?
- It seems that the model is effective by avoiding undesirable demonstrations, how to ensure the comprehensiveness of undesirable demonstrations?

[1] Wu Y H, Charoenphakdee N, Bao H, et al. Imitation learning from imperfect demonstration[C]//International Conference on Machine Learning. PMLR, 2019: 6818-6827.

[2] Wang Y, Xu C, Du B, et al. Learning to weight imperfect demonstrations[C]//International Conference on Machine Learning. PMLR, 2021: 10961-10970.

[3] Zhang S, Cao Z, Sadigh D, et al. Confidence-aware imitation learning from demonstrations with varying optimality[J]. Advances in Neural Information Processing Systems, 2021, 34: 12340-12350.

[4] Liu Q, Fu H, Tang K, et al. PN-GAIL: Leveraging Non-optimal Information from Imperfect Demonstrations[C]//The Thirteenth International Conference on Learning Representations.

**Ethical Concerns:**

["NO or VERY MINOR ethics concerns only"]

**Final Justification:**

My concerns have been resolved, and I will raise my score to 4.

**Limitations:**

- The limitations of the method have been outlined in the conclusion.

**Paper Formatting Concerns:**

No formatting issues found.

**Quality:**

3

**Strengths And Weaknesses:**

### Strengths
- Well-written paper with clear objectives. The authors have the intention to share the code.
- The question raised in the paper is reasonable and practical.
- Theoretical analysis is complete.
- The proposed method demonstrates good performance compared to the baselines.

### Weaknesses
- This approach seems to simply add the setting of undesirable demonstrations to other methods.
- As mentioned in the paper in Limitations, 'each undesirable trajectory may not be undesirable in its entirety', which may affect the final effect when used in practice.

---

> ### Author Rebuttal · Authors · 2025-07-30
>
> *We thank the reviewer for the insightful comments. Below, we provide our detailed responses to each of your questions and concerns.*
>
> ---
>
> > This approach seems to simply add the setting of undesirable demonstrations to other methods.
>
> We thank the reviewer for the feedback. We would like to emphasize that our work introduces a novel perspective by explicitly focusing on learning from undesirable behaviors—a setting that diverges significantly from conventional IL, which typically relies on expert or near-expert demonstrations. This shift presents a fundamentally new problem, requiring new training objectives and theoretical insights. Notably, one of our key and novel findings is that imitation learning can still succeed even when no expert demonstrations are explicitly identified in the dataset. We believe this opens an important and underexplored direction in the field, and represents a meaningful contribution to advancing the boundaries of IL research.
>
> > As mentioned in the paper in Limitations, 'each undesirable trajectory may not be undesirable in its entirety', which may affect the final effect when used in practice.
>
> We thank the reviewer for the insightful comment. Extending our approach to handle partially undesirable trajectories is indeed valuable, but it introduces significant challenges. Our method assumes each trajectory comes from a clearly defined policy—either desirable or undesirable—an assumption that breaks down when trajectories are mixed. Moreover, current benchmarks like D4RL lack segment-level labels needed to distinguish good and bad parts within a trajectory. Handling this would require additional supervision and methodological changes, which we consider a promising direction for future work beyond the scope of this paper.
>
> > If the probability of the demonstration appearing in undesirable demonstrations is less than the probability of appearing in unlabeled data, then is the optimization direction of Equation 4 reversed, and have you done any relevant experiments? For example, you can simply duplicate two copies of the undesirable demonstrations or add more to the unlabeled data.
>
> We thank the reviewer for the helpful comment. To clarify, our learning objective in Equation 4 does not involve the unlabeled dataset directly, so the optimization direction is theoretically unaffected even if the probability of a demonstration appearing in the undesirable dataset is lower than in the unlabeled dataset.
>
> Your point regarding adding more undesirable samples to the unlabeled data is well taken. To further investigate this, we conducted an additional experiment (see below) where we substantially increased the proportion of bad demonstrations in the unlabeled dataset to analyze how performance changes as its quality degrades.
>
> **Experimental description:** In this experiment, we evaluate two Mujoco tasks. We keep the same undesirable dataset $\mathcal{D}^{UN}$ as used in the main comparison in Section 5.2, consisting of 5 random and 5 medium trajectories. For the unlabeled dataset $\mathcal{D}^{MIX}$, we also retain the same 100 expert trajectories from Section 5.2, but vary the number of random and medium trajectories. Specifically, the unlabeled dataset has the format X:100, where X is the total number of random and medium trajectories combined (with X/2 random and X/2 medium), together with the 100 expert trajectories.
>
>
>
> **Cheetah**
>
> | | 1000:100 | 1500:100 | 2000:100 | 2500:100 | 3000:100 |
> |---|---|---|---|---|---|
> | SafeDICE | 3.1±0.9 | 5.1±5.2 | 5.2±4.0 | 4.3±6.1 | 2.4±3.2 |
> | DWBC | 2.2±0.0 | 2.2±0.0 | 2.2±0.0 | 2.2±0.0 | 2.2±0.0 |
> | UNIQ | 75.7±6.8 | 76.0±1.1 | 71.2±4.9 | 67.5±3.1 | 64.8±5.3 |
>
> **Ant**
>
> | | 1000:100 | 1500:100 | 2000:100 | 2500:100 | 3000:100 |
> |---|---|---|---|---|---|
> | SafeDICE | 4.4±2.6 | 5.8±7.8 | 5.6±3.7 | 4.7±4.1 | 5.3±5.8 |
> | DWBC | 84.6±12.4 | 84.9±1.9 | 84.4±3.8 | 83.4±4.3 | 84.3±6.8 |
> | UNIQ | 104.4±10.5 | 105.2±9.0 | 100.0±7.2 | 100.7±7.3 | 99.5±8.8 |
>
> > At present, the number of experts in the setup is relatively small. Is there an experiment with a larger number of experts, and will it be better than other methods?
>
> We thank the reviewer for the comment. To clarify, expert demonstrations are included only in the unlabeled dataset. As detailed in Section E.3 of the Appendix, we conducted an ablation study evaluating the performance of UNIQ and key baselines under varying amounts of expert data (from 50 to 300 demonstrations, with 300 considered a relatively large amount). The results show that UNIQ consistently outperforms the baselines across all settings.
>
> > Have you tried adding optimal demonstrations and undesirable demonstrations to train together, and will the results of UNIQ be better
>
> We thank the reviewer for the insightful comments. Our work is primarily focused on the setting where expert (or optimal) demonstrations are not available, and the algorithm is explicitly designed with this assumption in mind. Incorporating expert demonstrations directly into our learning objective (Equation 4) would substantially alter its structure and potentially lead to instability, as the learning signals from expert and undesirable data may conflict and drive the optimization in opposite directions. As such, we believe the current form of our algorithm is not well-suited for simultaneously learning from both expert and undesirable demonstrations. Addressing this scenario would likely require a fundamentally new algorithmic framework.
>
> That said, in settings where high-quality expert demonstrations are available, existing methods designed for learning from both expert and sub-optimal data—such as DemoDICE or DWBC—may be more appropriate.
>
> > The cost metric of the Car-Button in table 1 is much lower than the UNIQ on the LS-IQ method, but it shows that the UNIQ is the best, is it a recording error?
>
> Thank you for the comment. In this experiment, the LS-IQ policy achieves low cost but at the expense of an unacceptably low reward. In fact,  LS-IQ policy in fact does nothing, so the  cost is low, but it also fail to gain meaningful rewards. Therefore,  it is reasonable not to consider LS-IQ’s performance as best in this case. We will clarify this point in the updated version of the paper.
>
> > What are the advantages of the method proposed in this paper compared to confidence based imitation learning methods, such as 2IWIL, CAIL, WGAIL, PN-GAIL [1-4].
>
> We thank the reviewer for the comment. The key difference between our method and confidence-based approaches like 2IWIL, CAIL, WGAIL, and PN-GAIL is that those methods rely on per-sample confidence estimation and typically assume access to expert or near-expert demonstrations. In contrast, UNIQ is specifically designed for the more challenging setting where expert data is unavailable and only undesirable and unlabeled demonstrations are provided. This makes UNIQ more practical in scenarios where collecting high-quality expert data is infeasible.
>
> > Has the time spent in each method been recorded?
>
> We thank the reviewer for the comment. To clarify this, we compare the training speed of UNIQ to DWBC, SafeDICE, IPL, and LSIQ on a single task (Cheetah), using the same RTX 3090 GPU. The training speeds have been recorded as below:
>
> | algo | DWBC | SafeDICE | IPL | LSIQ | UNIQ |
> |---|---|---|---|---|---|
> | Walltime | ~150 mins | ~250 mins | ~90 mins | ~140 mins | ~30 mins |
>
> > It seems that the model is effective by avoiding undesirable demonstrations, how to ensure the comprehensiveness of undesirable demonstrations?
>
> We thank the reviewer for the thoughtful comment. While achieving full coverage of undesirable demonstrations is indeed challenging, our work is motivated by the practical observation that collecting undesirable data is often easier and less costly than collecting expert demonstrations. In real-world scenarios, such data can be obtained from logged failures or suboptimal policies, which typically span a sufficiently diverse set of poor actions. Based on this, we assume that the undesirable dataset provides adequate coverage of common failure modes. Moreover, our experiments—including those presented in the appendix—demonstrate that UNIQ remains robust across a variety of settings. That said, we also observe a decline in performance when the number of undesirable demonstrations is very low, which is expected and consistent with our assumptions.
>
> ---
>
> *We hope the above responses satisfactorily address your concerns. We will update the paper to include the additional experiments  and discussion described above. If you have any further comments or questions, we would be happy to address them.*

---

> > ### Comment · Reviewer_nyTV · 2025-08-05
> >
> > I appreciate the authors' response and have no further questions.

---

### Official Review · Reviewer_tHAr · 2025-07-02

**Clarity:** 3
**Significance:** 3
**Originality:** 3
**Rating:** 4
**Confidence:** 3

**Summary:**

The paper introduces UNIQ, an offline imitation learning framework for avoidance learning where the goal is to avoid undesirable behaviors based solely on negative demonstrations and unlabeled data, without relying on expert data. In contrast to traditional imitation learning, which learns by mimicking expert behavior, this method learns to avoid behaviors exhibited in low-quality demonstrations.

UNIQ formulates the problem using a MaxEnt inverse reinforcement learning objective, reformulated in the Q-space to enable stable optimization. The method combines Extreme Q-learning (to address non-convexity) with a novel occupancy correction mechanism that leverages unlabeled data in a statistically unbiased way. The authors demonstrate the approach on Mujoco and Safety-Gym benchmarks, where UNIQ outperforms several baselines, including SafeDICE and DWBC, in both return and constraint satisfaction.

**Questions:**

The discriminator estimating occupancy ratios plays a central role. How robust is this estimation under class imbalance or covariate shift between DUN and DMIX? Would the method degrade gracefully or fail catastrophically?

The authors note partial undesirability in demonstrations as a limitation. Could the method be extended to segment trajectories and learn from sub-trajectory labeling, possibly via importance weighting or causal inference?

While the paper claims minimal hyperparameter tuning, can the authors provide more details on how sensitive UNIQ is to choices like β, the regularization coefficient, or the discriminator architecture?

Have the authors considered applying this to high-dimensional observation spaces (e.g., images) or domains with long-horizon credit assignment challenges?

Although occupancy correction is intended to remove bias, is there any empirical observation on how performance changes with extreme noise or very low quality unlabeled data?

If the robustness of occupancy ratio estimation is shown to fail under modest noise, that would weaken the empirical claims.

A demonstration of some adaptation to partially undesirable trajectories would significantly increase the paper’s practical impact.

**Ethical Concerns:**

["NO or VERY MINOR ethics concerns only"]

**Final Justification:**

I believe my questions have been addressed, I confirm my score, and I have completed the mandatory acknowledgement.

**Limitations:**

The paper includes a discussion of limitations in Section 6.
The authors acknowledge:
_ the assumption of a single, fully undesirable dataset.
_ that some undesirable trajectories may contain good segments.
_ that more sophisticated use of sub-trajectory analysis could improve sample efficiency.
_ that learning from mistakes typically requires more data than learning from expert behavior.

One additional point that could be added: the method assumes availability of clearly labeled undesirable data, which may not always be available or well-defined in real-world applications. This could be mitigated by future work that incorporates human feedback or automatic anomaly detection.

**Paper Formatting Concerns:**

No concerns.

**Quality:**

3

**Strengths And Weaknesses:**

Quality:
+ The paper presents a rigorous and well-motivated formulation of the "learning from mistakes" problem.
+ Theoretical grounding is strong, with multiple propositions and a solid use of recent developments in inverse Q-learning and occupancy ratio estimation.
+ Experimental results are extensive and convincingly demonstrate the superiority of the proposed approach over strong baselines across diverse tasks.
- The performance relies on estimating occupancy ratios from a classifier trained on undesirable vs. unlabeled data. While standard, this method may be brittle in real-world scenarios, but this is not deeply analyzed.
- The practical aspects of scaling to more complex domains (e.g., larger action/state spaces, partial observability) are not explored.

Clarity
+ The writing is generally clear, and most derivations and concepts are well explained.
+ Illustrative examples (e.g., Figure 2) are helpful in clarifying the objective and behavior of the learning algorithm.
- Some sections (e.g., Section 4.3 on Extreme Q-learning) may be hard to parse without prior familiarity. The presentation could benefit from more intuitive explanations or graphical summaries.
- The distinction between SafeDICE and UNIQ in terms of practical behavior could be better emphasized.

Significance
+ The paper addresses an under-explored but practically important problem: how to learn safe or desirable behavior in the absence of expert data. The problem is relevant in real-world domains like autonomous driving or healthcare, where mistakes are more abundant than expert behavior.
+ The method achieves strong empirical results across multiple environments.
- While the use of unlabeled data is novel in this context, the assumptions about being able to isolate undesirable examples may limit applicability in domains without clear labels.

Originality
+ The approach is original in shifting the focus of imitation learning from expert-mimicking to anti-imitation of undesirable behavior.
+ The use of Extreme Q-learning to ensure convexity and the derivation of a novel cooperative training objective are significant innovations.
+ UNIQ’s unbiased use of unlabeled data via occupancy correction distinguishes it from prior work.
Some components (e.g., occupancy correction, WBC) are adapted from existing literature. The novelty is more in the integration and framing than in entirely new algorithms.

---

> ### Author Rebuttal · Authors · 2025-07-30
>
> *We thank the reviewer for the insightful comments. Below, we provide our detailed responses to each of your questions and concerns.*
>
> ---
>
> > The discriminator estimating occupancy ratios plays a central role. How robust is this estimation under class imbalance or covariate shift between DUN and DMIX? Would the method degrade gracefully or fail catastrophically?
> > If the robustness of occupancy ratio estimation is shown to fail under modest noise, that would weaken the empirical claims.
>
> We thank the reviewer for the insightful comment. It is indeed true that the estimation of occupancy ratios can be inaccurate when datasets are small or imbalanced. However, as shown in our experiments (especially those reported in appendix), despite the UN (undesirable) dataset typically having a low sample size, our UNIQ algorithm remains robust and effective.
>
> The key reason is that precise estimation of the occupancy ratio is not strictly necessary for learning a good policy. Specifically, in our main training objective (Equation 6), the discriminator is designed to assign a higher value to τ(s,a) when the state-action pair (s,a) appears more frequently in the UN dataset, and lower values otherwise. As a result, the objective encourages the learned Q-function and policy to assign lower rewards r^Q(s,a) to undesirable state-action pairs, thereby guiding the policy away from undesirable behaviors and toward more desirable ones.
>
> This phenomenon is consistent with prior findings in related works, such as GAIL and DICE-based algorithms, where learning remains effective even when occupancy ratios are estimated from low-sample datasets.
>
> > The authors note partial undesirability in demonstrations as a limitation. Could the method be extended to segment trajectories and learn from sub-trajectory labeling, possibly via importance weighting or causal inference?
>
> We thank the reviewer for the insightful comments. As mentioned in our paper, extending our approach to handle partially undesirable trajectories would introduce several new challenges:
> - **Violation of policy-level assumptions:** Our method relies on estimating occupancy measures for undesirable and mixed policies, assuming that each dataset is generated by a well-defined policy—either desirable or undesirable. When trajectories are only partially undesirable, this assumption no longer holds, as the generating policy cannot be clearly classified as good or bad. This ambiguity makes our current formulation unsuitable for such cases.
> - **Lack of segment-level labels:** Identifying which parts of a trajectory are desirable or undesirable is itself a challenging task. Existing benchmark datasets such as D4RL do not provide this level of granularity—they typically consist of full trajectories generated by either expert or non-expert policies, without annotations indicating which segments are good or bad. Addressing this would require additional supervision or expert feedback, as well as new methodological tools and thorough empirical validation.
>
> Given the complexity and scope of these challenges, we believe this extension falls outside the focus of our current work, but we agree it is a promising direction for future research.
>
> > While the paper claims minimal hyperparameter tuning, can the authors provide more details on how sensitive UNIQ is to choices like β, the regularization coefficient, or the discriminator architecture?
>
> We thank the reviewer for the comment. To address this, we conducted additional ablation studies to analyze the impact of the entropy regularization parameter β on the performance of UNIQ. The results are presented below. We note that β is a standard hyperparameter in maximum entropy frameworks, and its effects have been extensively studied in prior work. Our findings align with those results, showing that performance remains stable across a reasonable range of  values.
>
> For other standard hyperparameters—such as the regularization strength and the discriminator architecture—we adopted settings from prior work. These choices have been shown to be effective and well-validated,  representing strong defaults for our setting.
>
> | $\beta$ | 0.5 | 1 | 3 | 5 | 10 |
> |---|---|---|---|---|---|
> | Cheetah | 13.7±4.8 | 16.2±3.0 | 74.4±3.9 | **75.3±4.3** | 6.3±5.7 |
> | Ant | 90.8±20.0 | **104.3±1.6** | 57.2±19.3 | 26.5±4.2 | 24.6±8.2 |
> | Hopper | 7.3±3.9 | 4.0±1.8 | 43.4±20.9 | 73.4±16.8 | **75.1±11.0** |
> | Walker | 20.2±14.7 | 34.5±20.0 | 80.6±15.2 | **105.9±4.0** | 56.8±4.6 |
>
> > Have the authors considered applying this to high-dimensional observation spaces (e.g., images) or domains with long-horizon credit assignment challenges?
>
>  We thank the reviewer for the thoughtful comments. In this work, we have focused our experiments on widely used benchmark tasks from the D4RL suite, which are standard in the RL and IL communities. These tasks provide a solid foundation for evaluating and comparing the performance of our method against state-of-the-art baselines.
>
> We acknowledge that applying our approach to more complex settings—such as high-dimensional observation spaces (e.g., vision-based tasks) or domains with long-horizon credit assignment challenges—would be a valuable extension. However, these settings often require significant additional engineering effort (e.g., integrating convolutional architectures or temporal abstraction mechanisms) and computational resources for robust experimentation and validation. We consider this an important and promising direction for future work.
>
> > Although occupancy correction is intended to remove bias, is there any empirical observation on how performance changes with extreme noise or very low quality unlabeled data?
>
> We thank the reviewer for the insightful comment. To further address this point, we conducted additional experiments to evaluate the performance of our algorithm under conditions where the unlabeled dataset is of extremely low quality. Specifically, we varied the number of suboptimal (bad) demonstrations within the unlabeled set, thereby progressively degrading its overall quality. As expected, we observed a noticeable performance degradation as more bad demonstrations were added to the unlabeled dataset. When the overall quality of the unlabeled data became extremely poor, the algorithm's performance dropped significantly. This outcome is reasonable, as the learning signal becomes increasingly noisy and less informative in such scenarios.
>
> **Experimental description:** In this experiment, we evaluate two Mujoco tasks. We keep the same undesirable dataset $\mathcal{D}^{UN}$ as used in the main comparison in Section 5.2, consisting of 5 random and 5 medium trajectories. For the unlabeled dataset $\mathcal{D}^{MIX}$, we also retain the same 100 expert trajectories from Section 5.2, but vary the number of random and medium trajectories. Specifically, the unlabeled dataset has the format X:100, where X is the total number of random and medium trajectories combined (with X/2 random and X/2 medium), together with the 100 expert trajectories.
>
>
>
> **Cheetah**
>
> | | 1000:100 | 1500:100 | 2000:100 | 2500:100 | 3000:100 |
> |---|---|---|---|---|---|
> | SafeDICE | 3.1±0.9 | 5.1±5.2 | 5.2±4.0 | 4.3±6.1 | 2.4±3.2 |
> | DWBC | 2.2±0.0 | 2.2±0.0 | 2.2±0.0 | 2.2±0.0 | 2.2±0.0 |
> | UNIQ | 75.7±6.8 | 76.0±1.1 | 71.2±4.9 | 67.5±3.1 | 64.8±5.3 |
>
> **Ant**
>
> | | 1000:100 | 1500:100 | 2000:100 | 2500:100 | 3000:100 |
> |---|---|---|---|---|---|
> | SafeDICE | 4.4±2.6 | 5.8±7.8 | 5.6±3.7 | 4.7±4.1 | 5.3±5.8 |
> | DWBC | 84.6±12.4 | 84.9±1.9 | 84.4±3.8 | 83.4±4.3 | 84.3±6.8 |
> | UNIQ | 104.4±10.5 | 105.2±9.0 | 100.0±7.2 | 100.7±7.3 | 99.5±8.8 |
>
>
> ---
>
> *We hope the above responses satisfactorily address your concerns. We will update the paper to include the additional experiments  and discussion described above. If you have any further comments or questions, we would be happy to address them*

---

> > ### Comment · Reviewer_tHAr · 2025-08-06
> >
> > Thank you for the detailed rebuttal. The authors have adequately addressed my concerns, and I appreciate the clarifications provided.

---

### Decision · Program_Chairs · 2025-09-17

**Decision:**

Accept (poster)

**Comment:**

The submitted paper explores learning avoidance behavior in offline imitation learning without access to expert demonstrations, i.e., avoiding undesirable behavior learned from negative examples. The authors propose a novel training objective (UNIQ) based on the maximum entropy principle and reformulate the problem as a cooperative inverse Q-learning task. They also introduce a method to incorporate unlabeled data for unbiased training. Experimental results on benchmark environments demonstrate that their approach outperforms state-of-the-art baselines in several settings.

**Strengths of the Paper**
* Problem setting: The paper addresses a practically relevant and under-explored problem, i.e., learning from undesirable demonstrations without expert data. This can be particularly relevant in safety-critical domains like autonomous driving and healthcare.
* Theoretical framework: The paper provides a sensible reformulation of the MaxEnt inverse reinforcement learning objective in Q-space, along with a novel occupancy correction mechanism.
* Empirical evaluation: Experiments on various benchmarks (Mujoco, Safety-Gym) demonstrate that UNIQ can achieve superior performance compared to baselines, particularly in safety-constrained settings. The authors also provide detailed ablation studies and additional experiments in the rebuttal, which when added to the paper, make the experimental evaluation solid and “complete”.

**Weaknesses of the Paper**
* Novelty: Some reviewers noted that the method primarily builds on existing ideas (e.g., IQ-Learn, Extreme Q-learning, occupancy ratio estimation) and flips the objective of IQ-Learn. While the integration is novel, the individual components are not entirely new.
* Dataset assumptions: The method assumes that undesirable demonstrations are clearly labeled and that the unlabeled dataset contains a mix of good and bad behaviors. This assumption may not hold in real-world scenarios where such clear distinctions are unavailable.
* Robustness concerns: The reliance on occupancy ratio estimation via a binary classifier could result in limited robustness. The authors provided additional experiments to address emphasize robustness (and that exact estimation might not be necessary), further analysis would strengthen the paper.


** Discussion**
Both reviewers and authors actively engaged in the discussion phase. In particular, the following central points were discussed:
* Realism of assumptions: One reviewer raised concerns about the assumption of clearly labeled undesirable data and the conflation of suboptimal and undesirable behaviors. Another reviewer argued that the assumption of negative samples is realistic in safety-critical settings, where undesirable behaviors (e.g., collisions) are easier to classify than expert behaviors. The authors clarified in response that their method is designed for settings where undesirable data is easier to collect than expert data. They acknowledged the limitations of their assumptions and suggested future work on sub-trajectory labeling and human feedback (which introduces additional challenges).
* Occupancy ratio estimation: One reviewer expressed concerns about the accuracy of occupancy ratio estimation and its impact on policy learning. The authors conducted additional experiments to evaluate the robustness of the discriminator and showed that precise ratio estimation is not strictly necessary for effective learning.
* Baselines and comparisons: One reviewer suggested an alternative adaptation of DWBC for the proposed problem setting. The authors implemented this suggestion and compared it to their adapted version of DWBC. However, both versions underperform compared to UNIQ.
* Data composition and scaling: One reviewer suggested experiments with more diverse data compositions. The authors conducted these experiments and found that their proposed UNIQ (as well as DWBC-based methods) struggles.
* Impact of extreme Q-learning: One reviewer suggested an ablation study to evaluate the impact of extreme Q-learning. The authors conducted this ablation and provided results showing that extreme Q-learning achieves better performance compared to other Q-function update strategies (e.g., IQL, SQL).

**Recommendation**
The paper addresses a novel and practically important problem, provides a well-motivated and theoretically grounded solution, and demonstrates strong empirical results across diverse benchmarks. While some concerns about robustness and dataset assumptions remain, the authors have adequately addressed these during the rebuttal period through additional experiments and clarifications. UNIQ’s focus on learning from undesirable behaviors without expert data represents a meaningful contribution to the field of imitation learning. Hence, in line with the recommendation of all reviewers, I recommend the acceptance of the paper.